# DiffFlow: A Unified SDE for Score-Based Diffusion Models and Generative Adversarial Networks

## Abstract

Generative models can be categorized into two types: i) explicit generative models that define explicit density forms and allow exact likelihood inference, such as score-based diffusion models (SDMs) and normalizing flows; ii) implicit generative models that directly learn a transformation from the prior to the data distribution, such as generative adversarial nets (GANs). While these two types of models have shown great success, they suffer from respective limitations that hinder them from achieving fast sampling and high sample quality simultaneously. In this paper, we propose a unified theoretic framework for SDMs and vanilla GANs (Goodfellow et al. (2020)). We mainly show that: i) the generation process of both SDMs and GANs can be described as a novel SDE named **D**iscriminator Deno**i**sing Di**ff**usion **Flow** (**DiffFlow**), where the drift can be determined by some weighted combinations of scores of the real data and the generated data; ii) By adjusting the relative weights between different score terms, we can obtain a smooth transition between SDMs and GANs while the marginal distribution of the SDE remains invariant to the change of the weights; iii) we prove the asymptotic and non-asymptotic convergence of the continuous SDE dynamics of DiffFlow by some weak isoperimetry of the smoothed target distribution; iv) under our unified theoretic framework, we introduce several instantiations of DiffFlow that incorporate some recently proposed hybrid algorithms of GAN and diffusion models, for instance, the TDPM (Zheng et al., 2022) as a special case. Our framework unifies GANs and SDMs into a continuous spectrum. Hence, it offers the potential to design new generative learning algorithms that could achieve a flexible trade-off between high sample quality and fast sampling speed beyond existing GAN- and/or SDM-like algorithms.

## 1 Introduction

Generative modeling is a fundamental task in machine learning: given finite i.i.d. observations from an unknown target distribution, the goal is to learn a parametrized model that transforms a known prior distribution (e.g., Gaussian noise) to a distribution that is "close" to the unknown target distribution. In the past decade, we have witnessed rapid developments of a plethora of deep generative models (i.e., generative modeling based on deep neural networks): starting from VAEs (Kingma & Welling, 2013), GANs (Goodfellow et al., 2020), Normalizing Flows (Rezende & Mohamed, 2015), and more recently, score-based diffusion models (SDMs) (Sohl-Dickstein et al., 2015; Ho et al., 2020). These deep generative models have demonstrated remarkable capabilities in modeling high-dimensional distributions, which pose challenges for traditional "shallow" generative models such as Gaussian Mixture Models.

Despite the existence of a large family of deep generative models, they can be categorized into two groups based on sampling and likelihood inference. The first one is explicit generative models, which define explicit density forms and enable exact likelihood inference (Huang et al., 2021; Song et al., 2021a; Kingma et al., 2021) using the well-known Feynman-Kac formula (Karatzas et al., 1991). Score-based diffusion models and normalizing flows are typical examples of explicit generative models. Another group is implicit generative models such as GANs directly learn a transformation from a noise prior to the data distribution, making the closed-form density of the

learned distribution intractable. In this work, we focus on GANs as representatives of implicit generative models and SDMs as representatives of explicit generative models, as they exhibit superior performance within their respective classes of generative models.

GANs are trained through a minimax game between the generator network and the discriminator network. They have been one of the pioneering implicit generative models that have dominated the field of image generation for many years. The sampling process of GANs is fast since it only requires a single pass through the generator network to transform the noise vector into a data vector. However, GANs suffer from training instability due to the nonconvex-nonconcave objective function, and the quality of generated samples is often inferior compared to the current state-of-the-art score-based diffusion models (Dhariwal & Nichol, 2021). In contrast to GANs, SDMs achieve high-quality image generation without adversarial training. SDMs (Song et al., 2021b) are explicit generative models that define a forward diffusion process, iteratively transforming the data into random noise, and the learning objective is to reverse this forward diffusion process using a reverse denoising process. The relationship between denoising and score matching is well-known in the literature (Hyvärinen & Dayan, 2005), which explains the term "score" in SDMs. However, the iterative nature of SDMs makes both the training and sampling processes significantly slower compared to GANs.

While significant progress has been made in the individual fields of GANs and diffusion models, there has been limited research on linking and studying the relationship between these two approaches. In this work, our objective is to address the following research question:

*Can we develop a unified theoretical framework for GANs and SDMs*
*that allows for a flexible trade-off between high sample quality and fast sampling speed?*

The goal of this paper is to provide a positive response to the aforementioned question. Our contributions can be summarized into four parts:

1. Our key observation is that the generation process of both SDMs and GANs can be described by a novel stochastic differential equation (SDE) named **D**iscriminator Deno**i**sing Dif**f**usion **Flow** (**DiffFlow**). The drift term of DiffFlow comprises a weighted combination of scores from the current marginal distribution $p_t(x)$ at time $t$ and the target distribution $q(x)$.

2. By carefully adjusting the weights of the scores in the drift term, we achieve a smooth transition between GANs and SDMs. Here, "smooth" implies that the marginal distribution $p_t(x)$ remains unchanged during the weight adjustments from GANs to SDMs. We refer to this property as the "Marginal Preserving Property," which we rigorously define in Section 3.4.3.

3. We provide asymptotic and non-asymptotic convergence analyses of the dynamics of the proposed SDE under certain weak isoperimetry properties of the smoothed target distribution. Additionally, we derive a training objective that guarantees maximal likelihood.

4. Finally, within our unified SDE framework, we present several instantiations of SDE dynamics that encompass various recently proposed empirical hybrid algorithms of GANs and diffusion models as special cases. One notable example is the truncated probabilistic diffusion models (Zheng et al., 2022), which employ a two-stage algorithm: GANs learn a one-step mapping from pure noise to a noisy target distribution, and the remaining chain is implemented by standard diffusion steps. Our unified framework offers the potential to design new generative learning algorithms that allow for a flexible trade-off between high sample quality and fast sampling speed.

The remainder of this paper is organized as follows: Section 2 provides background information for general readers. Our unified SDE framework is presented in Section 3, and the asymptotic convergence analysis is discussed in Section 4. Section 5 covers the related work. Due to space constraints, we defer all proofs and some theoretical results to the Appendix. Specifically, the non-asymptotic convergence analysis of DiffFlow is deferred to Appendix H, and the derivation of the maximal likelihood training scheme is deferred to Appendix I. Appendix J further discusses new algorithms generated by our framework and the analytic continuation of the noise coefficients. In Appendix K,

we explore potential design spaces under our framework and their relationship to existing hybrid algorithms of GANs and diffusion models, such as TDPM proposed by (Zheng et al., 2022). Finally, we conclude the paper in Section 6.

## 2 BACKGROUND

### 2.1 SCORE-BASED DIFFUSION MODELS

SDMs are a type of generative models trained by denoising samples corrupted by various levels of noise. The generation process involves sampling vectors from pure noise and progressively denoising them to generate images. Song et al. (2021b) formally describes these processes using a forward diffusion stochastic differential equation (SDE) and a reverse denoising SDE.

Specifically, let the data distribution be denoted by $q(x)$. By sampling a particle $X_0 \sim q(x)$, the forward diffusion process $\{X_t\}_{t \in [0,T]}$ is defined by the following SDE:

$$dX_t = f(X_t, t)dt + g(t)dW_t, \tag{1}$$

where $T > 0$ is a fixed constant, $f(\cdot, \cdot) : \mathbb{R}^k \times [0, T] \to \mathbb{R}^k$ is the drift coefficient, $g(\cdot) : [0, T] \to \mathbb{R}_{\geq 0}$ is a predefined diffusion noise scheduler, and $\{W_t\}_{t \in [0,T]}$ is the standard Brownian motion in $\mathbb{R}^k$. If we denote the probability density of $X_t$ by $p_t(x)$, our goal is to have the distribution of $p_T(x)$ be close to a tractable Gaussian distribution $\pi(x)$. Setting $f(X_t, t) \equiv 0$ and $g(t) = \sqrt{2t}$, Karras et al. (2022) yields $p_t(x) = q(x) \circledast \mathcal{N}(0, t^2 I) := q(x; t)$, where $\circledast$ denotes the convolution operation and $q(x, T) \approx \pi(x) = \mathcal{N}(0, T^2 I)$.

The reverse process, as defined by Song et al. (2021b), involves sampling an initial particle from $X_0 \sim \pi(x) \approx p_T(x)$. The reverse denoising process $\{X_t\}_{t \in [0,T]}$ is then defined by the following SDE:

$$dX_t = \left[ g^2(T - t) \nabla \log p_{T-t}(X_t) - f(X_t, T - t) \right] dt + g(T - t) dW_t. \tag{2}$$

It is worth mentioning that this denoising process is a trivial time-reversal of the original reverse process defined by Song et al. (2021b) and also appears in previous work (Huang et al., 2021) for notational simplicity.

In the denoising process[1], the critical term to be estimated is $\nabla \log p_{T-t}(x)$, which represents the score function of the forward process at time $T-t$. $p_{T-t}(x)$ is often a noise-corrupted version of the target distribution $q(x)$. For example, as discussed earlier, if we set $f(X_t, t) \equiv 0$ and $g(t) = \sqrt{2t}$, then $\nabla \log p_{T-t}(x)$ becomes $\nabla \log q(x, T - t)$, which can be estimated through denoising score matching (Song et al., 2021b; Karras et al., 2022) using a time-indexed network $s_\theta(x, t)$ (Ho et al., 2020).

### 2.2 GENERATIVE ADVERSARIAL NETWORKS

In 2014, Goodfellow et al. (2020) introduced the seminal work on generative adversarial nets (GANs). The training dynamics of GANs can be formulated as a minimax game between a discriminator network $d_{\theta_D}(\cdot) : \mathcal{X} \to [0, 1]$ and a generator network $G_{\theta_G}(\cdot) : \mathcal{Z} \to \mathcal{X}$. Intuitively, the discriminator network is trained to classify images as fake or real, while the generator network is trained to produce images from noise that "fool" the discriminator. This alternating procedure for training the generator and discriminator can be formulated as a minimax game:

$$\min_{\theta_G} \max_{\theta_D} \mathbb{E}_{x \sim q(x)}[\log d_{\theta_D}(x)] + \mathbb{E}_{z \sim \pi(z)}[\log(1 - d_{\theta_D}(G_{\theta_G}(z)))], \tag{3}$$

where $z \sim \pi(z)$ is sampled from Gaussian noise. The training dynamics of GANs are unstable due to the high non-convexity of the generator and discriminator, which hinders the existence and uniqueness of the equilibrium in the minimax objective (Farnia & Ozdaglar (2020)).

Despite significant progress in the fields of diffusion models and GANs in recent years, little is known about the connection between them. In the next section, we will provide a general framework that unifies GANs and diffusion models, demonstrating that the generation dynamics of GANs and diffusion models can be recovered as a special case of our general framework.

---

[1] We omit the word "reverse" since this process is the time-reversal of the original reverse process.

## 3 GENERAL FRAMEWORK

To establish a unified stochastic differential equation (SDE) framework for GANs and SDMs, we propose disregarding the terminologies "forward process" and "reverse process" commonly used in diffusion models literature. Our objective is to construct a learnable (potentially stochastic) process $\{X_t\}_{t\in[0,T]}$ indexed by continuous time variable $0 \leq t \leq T$, such that $X_0 \sim \pi(x)$, where $\pi(x)$ is a known Gaussian distribution, and $X_T \sim p_T(x)$, which is "close" to the target distribution $q(x)$. The measure of closeness can be defined by a divergence or metric, on which we later will provide details.

Specifically, given $X_0 \sim \pi(x)$ sampled from the noise distribution, we consider the following evolution equation for $X_t \sim p_t(x)$, where $t \in [0, T]$ and $T > 0$:

$$dX_t = \left[f(X_t, t) + \beta(t, X_t)\nabla \log \frac{q(u(t)X_t; \sigma(t))}{p_t(X_t)} + \frac{g^2(t)}{2}\nabla \log p_t(X_t)\right] dt$$
$$+ \sqrt{g^2(t) - \lambda^2(t)}dW_t. \quad (4)$$

Here, $f(\cdot, \cdot) : \mathbb{R}^k \times [0, T] \to \mathbb{R}^k$, $\beta(\cdot, \cdot) : \mathbb{R}^k \times [0, T] \to \mathbb{R}_{\geq 0}$, and $u(\cdot), \sigma(\cdot), g(\cdot), \lambda(\cdot) : [0, T] \to \mathbb{R}_{\geq 0}$ are predefined scaling functions.

For ease of presentation, we refer to the above SDE as the **D**iscriminator Deno**i**sing Di**ff**usion **Flow** (**DiffFlow**). At first glance, the physical interpretation behind DiffFlow may not be immediately apparent. However, we will provide a detailed explanation of how the name DiffFlow is derived and how it unifies GANs and SDMs in subsequent subsections. By carefully tuning the scaling functions, we can recover the dynamics of GANs and SDMs respectively, while keeping the marginal distributions $p_t(x)$ unchanged. Furthermore, we will demonstrate how DiffFlow unifies a broader "continuous" spectrum of generative models, where GANs and diffusion models represent specific cases of DiffFlow with specialized scaling functions.

### 3.1 SINGLE NOISE-LEVEL LANGEVIN DYNAMICS

Let us begin with the simplest case: the single-noise level Langevin dynamics. In DiffFlow, if we set $u(t) \equiv 1$, $f(X_t, t) \equiv 0$, $\lambda(t) \equiv 0$, $\beta(t, X_t) \equiv \beta(t)$, $g(t) \equiv \sqrt{2\beta(t)}$, and $\sigma(t) \equiv \sigma_0$ for some fixed $\sigma_0 \geq 0$, then we obtain

$$dX_t = \beta(t)\nabla \log q(X_t; \sigma_0)dt + \sqrt{2\beta(t)}dW_t, \quad (5)$$

which corresponds to the classical Langevin algorithm. However, as pointed out by Song & Ermon (2019; 2020), the single noise level Langevin algorithm suffers from slow mixing time due to separate regions of the data manifold. Consequently, it is unable to model complex high-dimensional data distributions and fails to learn and generate meaningful features on datasets like MNIST (Song & Ermon, 2019). Therefore, it is reasonable to introduce perturbations with multiple noise levels and perform multiple noise-level Langevin dynamics, such as the Variance Explosion SDE with Corrector-Only Sampling (NCSN) (Song et al., 2021b).

### 3.2 SCORE-BASED DIFFUSION MODELS

#### 3.2.1 VARIANCE EXPLOSION SDE

Without loss of generality, we consider the Variance Explosion (VE) SDE adopted by Karras et al. (2022). DiffFlow can be easily adapted to other VE SDEs by simply changing the noise schedule $\sigma(t)$ and scaling $\beta(t, X_t)$. For instance, setting $u(t) \equiv 1$, $f(X_t, t) \equiv 0$, $\beta(t, X_t) \equiv 2(T - t)$, $\lambda(t) \equiv \sqrt{\beta(t)} = \sqrt{T - t}$, $g(t) \equiv \sqrt{2\beta(t)} = 2\sqrt{T - t}$, and $\sigma(t) \equiv T - t$, we obtain

$$dX_t = 2(T - t)\nabla \log q(X_t; T - t)dt + \sqrt{2(T - t)}dW_t, \quad (6)$$

which corresponds to the VE SDE from Karras et al. (2022). In the case of a general VE SDE as described by Song et al. (2021b), the denoising process is given by

$$dX_t = \frac{d[\sigma^2(T - t)]}{dt}\nabla \log q\left(X_t; \sqrt{\sigma^2(T - t) - \sigma^2(0)}\right) + \sqrt{\frac{d[\sigma^2(T - t)]}{dt}}dW_t, \quad (7)$$

This can be obtained by setting $u(t) \equiv 1$, $f(X_t, t) \equiv 0$, $\beta(t, X_t) \equiv \frac{d[\sigma^2(T-t)]}{dt}$, $\lambda(t) \equiv \sqrt{\beta(t, X_t)} = \sqrt{\frac{d[\sigma^2(T-t)]}{dt}}$, $g(t) \equiv \sqrt{2\beta(t, X_t)} = \sqrt{2\frac{d[\sigma^2(T-t)]}{dt}}$, and $\sigma(t) \equiv \sqrt{\sigma^2(T-t) - \sigma^2(0)}$ in DiffFlow.

### 3.2.2 VARIANCE PRESERVING SDE

Similar to the previous analysis, for the Variance Preserving (VP) SDE described by Song et al. (2021b), the denoising process is given by

$$dX_t = \left[\beta(T-t)\nabla \log p_{T-t}(X_t) + \frac{1}{2}\beta(T-t)X_t\right]dt + \sqrt{\beta(T-t)}dW_t. \tag{8}$$

According to the marginal distribution of VP SDE derived in Appendix A, this can be obtained by setting $u(t) \equiv \exp\left(\frac{1}{2}\int_0^{T-t}\beta(s)ds\right)$, $f(X_t, t) \equiv \frac{1}{2}\beta(T-t)X_t$, $\beta(t, X_t) \equiv \beta(T-t)$, $\lambda(t) \equiv \sqrt{\beta(T-t)}$, $g(t) \equiv \sqrt{2\beta(T-t)}$, and $\sigma(t) \equiv 1 - \exp\left(-\int_0^{T-t}\beta(s)ds\right)$ in DiffFlow. A similar procedure can demonstrate that the sub-VP SDE proposed by Song et al. (2021b) also lies within the framework with specialized scaling functions, which we omit the derivations here for brevity.

### 3.2.3 DIFFUSION ODE FLOW

Similar to the previous analysis, for the diffusion ODE corresponding to the VE SDE by Song et al. (2021b), the denoising process is given by

$$dX_t = \frac{1}{2}\frac{d[\sigma^2(T-t)]}{dt}\nabla \log q\left(X_t; \sqrt{\sigma^2(T-t) - \sigma^2(0)}\right). \tag{9}$$

This can be obtained by setting $u(t) \equiv 1$, $f(X_t, t) \equiv 0$, $\beta(t, X_t) \equiv \frac{1}{2}\frac{d[\sigma^2(T-t)]}{dt}$, $\lambda(t) \equiv \sqrt{2\beta(t, X_t)} = \sqrt{2\frac{d[\sigma^2(T-t)]}{dt}}$, $g(t) \equiv \sqrt{2\beta(t, X_t)} = \sqrt{2\frac{d[\sigma^2(T-t)]}{dt}}$, and $\sigma(t) \equiv \sqrt{\sigma^2(T-t) - \sigma^2(0)}$ in DiffFlow. The ODEs corresponding to VP SDEs and sub-VP SDEs can be obtained from DiffFlow by specializing the scaling functions using a similar procedure. However, we omit the derivations here for brevity.

### 3.3 GENERATIVE ADVERSARIAL NETWORKS

To start with the simplest case, let us demonstrate how DiffFlow recovers the training dynamics of the vanilla GAN (Goodfellow et al., 2020). First, we set $u(t) \equiv 1$, $f(X_t, t) \equiv 0$, $\lambda(t) \equiv 0$, $g(t) \equiv 0$, and $\sigma(t) \equiv \sigma_0 \geq 0$. It is worth noting that $\sigma_0$ is typically set to a small positive constant to ensure the smoothness of the generator's gradient. With these settings, DiffFlow reduces to the following DiffODE:

$$dX_t = \left[\beta(t, X_t)\nabla \log \frac{q(X_t; \sigma_0)}{p_t(X_t)}\right]dt. \tag{10}$$

Next, we demonstrate that by coarsely approximating the dynamics of this ODE using a generator network with specialized $\beta(t, X_t)$, one can recover the dynamics of the Vanilla GAN.

The critical term to be estimated in the DiffODE is $\nabla \log \frac{q(X_t; \sigma_0)}{p_t(X_t)}$, which represents the gradient field of the classifier between the real data and the generated data at time $t$. This term can be estimated by computing gradients with respect to the logistic classifier defined as follows:

$$D_t(x) := \log \frac{q(x; \sigma_0)}{p_t(x)} = \arg\min_D \left[\mathbb{E}_{x \sim q(x; \sigma_0)}\log\left(1 + e^{-D(x)}\right) + \mathbb{E}_{x \sim p_t(x)}\log\left(1 + e^{D(x)}\right)\right].$$

We can then update the samples using the following equation:

$$X_{t+1} = X_t + \eta_t\beta(t, X_t)\nabla D_t(X_t), \tag{11}$$

where $\eta_t > 0$ represents the discretization step size.

It is well established in the existing literature (Gao et al., 2019; Yi et al., 2023) that the vanilla GAN or non-saturating GAN dynamics can be exactly recovered by one-step distillation of the discriminator-guided particle dynamics, i.e., the aforementioned DiffODE. The equivalence between the vanilla GAN and discriminator-guided dynamics GANs can be obtained by rescaling the gradient of the least square distillation objective. For the sake of completeness and due to space limitations, we provide detailed discussions on the equivalence between DiffODE and vanilla GANs in Appendix B, as well as several improvements of vanilla GANs from the perspective of DiffODE in Appendix C.

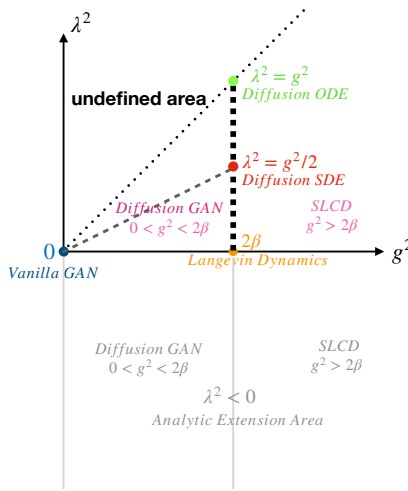

Figure 1: A Continuous Spectrum of Generative Models (Best Viewed in Color).

### 3.4 A UNIFIED SDE FRAMEWORK

In the previous sections, we have demonstrated that by customizing the scaling functions of DiffFlow, we can recover the dynamics of the single-noise Langevin algorithm, diffusion models, and GANs. However, DiffFlow provides a broader continuous spectrum of generative models, with GANs and SDMs representing specific corner cases on this spectrum, as depicted in Figure 1.

#### 3.4.1 DIFFFLOW DECOMPOSITION

Recall that the dynamics of DiffFlow can be described by

$$dX_t = \left[ \underbrace{f(X_t, t)}_{Regularization} + \underbrace{\beta(t)\nabla \log \frac{q(u(t)X_t; \sigma(t))}{p_t(X_t)}}_{Discriminator} + \underbrace{\frac{g^2(t)}{2}\nabla \log p_t(X_t)}_{Denoising} \right] dt$$

$$+ \underbrace{\sqrt{g^2(t) + \widetilde{\lambda}^2(t)}dW_t}_{Diffusion}, \quad (12)$$

where we assign names to each term based on their physical interpretation in particle evolutions: the first term $f(X_t, t)$, acting similarly to weight decay when $f(X_t, t) = c\|X_t\|_2^2$ for some constant $c > 0$, represents *regularization*; the second term, named the *discriminator*, corresponds to the gradient of the classifier between target data and real data; the third term, *denoising*, removes Gaussian noise with standard deviation $g(t)$ according to the Kolmogorov Forward Equation; and the last term is the *diffusion*.

While the physical meanings of the aforementioned terms are evident, explaining the continuous evolution of models between GANs and SDMs through the scaling functions $g(\cdot)$ and $\lambda(\cdot)$ is challenging. However, it is worth noting that when $g^2(t) \leq 2\beta(t)$, the DiffFlow equation can be alternatively written as:

$$dX_t = \left[ \frac{g^2(t)}{2}\nabla \log q(u(t)X_t; \sigma(t)) \right] dt + \sqrt{g^2(t) - \lambda^2(t)}dW_t$$

$$+ \left[ \left( \beta(t) - \frac{g^2(t)}{2} \right) \nabla \log \frac{q(u(t)X_t; \sigma(t))}{p_t(X_t)} + f(X_t, t) \right] dt.$$

This decomposition implies that when $g^2(t) \leq 2\beta(t)$, DiffFlow can be seen as a mixed particle dynamics between GANs and SDMs, where the relative mixture weight is controlled by $g(t)$ and

$\beta(t)$. If we fix $\beta(t)$ and increase $g(t)$ to $\sqrt{2\beta(t)}$, the dynamics of GANs would vanish. In the limit where $g(t) = \sqrt{2\beta(t)}$ and $\lambda(t) \equiv 0$, DiffFlow reduces to the pure Langevin algorithm. On the other hand, in the limit where $g(t) \equiv 0$ and $\lambda(t) \equiv 0$, DiffFlow reduces to the pure GANs algorithm.

Regarding the evolution from the pure Langevin algorithm to diffusion SDE models, we need to increase $\lambda(t)$ from 0 to $g(t)/\sqrt{2}$ to match the stochasticity of VP/VE SDE. If we further increase $\lambda(t)$ to $g(t)$, we obtain the diffusion ODE (Song et al., 2021b).

### 3.4.2 DIFFUSION-GAN: A UNIFIED ALGORITHM

Notice that when $0 < g(t) < \sqrt{2\beta(t)}$, DiffFlow exhibits a mixed particle dynamics of SDMs and GANs, which we refer to as Diffusion-GANs:

$$dX_t = \left[ \frac{g^2(t)}{2} \nabla \log q(u(t)X_t; \sigma(t)) \right] dt + \sqrt{g^2(t) - \lambda^2(t)} dW_t$$
$$+ \left[ \left( \beta(t) - \frac{g^2(t)}{2} \right) \nabla \log \frac{q(u(t)X_t; \sigma(t))}{p_t(X_t)} + f(X_t, t) \right] dt.$$

Under the DiffFlow framework, one can implement Diffusion-GANs by learning a time-indexed discriminator using logistic regression $d_{\theta_t^*}(x, t) \approx \log \frac{q(u(t)X_t; \sigma(t))}{p_t(X_t)}$ and a score network using score matching $s_{\theta'_*}(x, t) \approx \nabla \log q(u(t)X_t; \sigma(t))$. The sampling process is then defined by discretizing the following SDE within the interval $t \in [0, T]$:

$$dX_t = \left[ \frac{g^2(t)}{2} s_{\theta'_*}(x, t) \right] dt + \sqrt{g^2(t) - \lambda^2(t)} dW_t + \left[ \left( \beta(t) - \frac{g^2(t)}{2} \right) \nabla d_{\theta_t^*}(x, t) + f(X_t, t) \right] dt.$$

It is important to tune $\lambda(t)$ to achieve an appropriate level of stochasticity. The optimal noise level for generating high-quality samples remains an open problem both empirically and theoretically.

### 3.4.3 MARGINAL PRESERVING PROPERTY

One may argue that the current framework of DiffFlow is simply a combination of particle dynamics from GANs and SDMs, where $g(\cdot)$ serves as an interpolation weight between them. It is well-known in the literature that both GANs (Gao et al., 2019; Yi et al., 2023) and SDMs (Song et al., 2021b) can be modeled as particle dynamics using different ordinary or stochastic differential equations.

However, we want to emphasize that despite GANs and SDMs being modeled as ODEs/SDEs with different drift and diffusion terms, the underlying dynamics of these differential equations are distinct: even with the same initial particle configurations, the path measure deviates significantly between GANs and SDMs. Therefore, a simple linear interpolation between the dynamics of GANs and SDMs lacks both theoretical and practical motivations. We require a sophisticated design of each term in the differential equations to align the marginal distributions of the different classes of generative models represented by SDMs and GANs. This work presents a well-designed unified SDE called "DiffFlow" for GANs and SDMs, enabling flexible interpolations between them by adjusting the scaling functions in the SDE. Furthermore, we demonstrate in the following proposition that the interpolation is "smooth" from GANs to SDMs: the marginal distribution $p_t(x)$ for all $t \geq 0$ remains invariant to the interpolation factor $g(\cdot)$. This proposition is a direct consequence of the Fokker-Planck equation (Jordan et al. (1998)).

**Proposition 1** (Marginal Preserving Property, Proved in Appendix D). *The marginal distribution $p_t(x)$ of DiffFlow, given by*

$$dX_t = \left[ f(X_t, t) + \beta(t) \nabla \log \frac{q(u(t)X_t; \sigma(t))}{p_t(X_t)} + \frac{g^2(t)}{2} \nabla \log p_t(X_t) \right] dt + \sqrt{g^2(t) - \lambda^2(t)} dW_t$$

*remains invariant with respect to $g(\cdot)$.*

The Marginal Preserving Property implies that as we increase $g(\cdot)$, DiffFlow transitions smoothly from GANs to SDMs and further to SLCD (see Appendix J for details), as illustrated in Figure 1, while the marginal distribution $p_t(x)$ remains unchanged. Another important factor, $\lambda(\cdot)$, is used to control the stochasticity of DiffFlow, aligning the noise level among Langevin Dynamics, diffusion SDEs, and diffusion ODEs.

## 4 CONVERGENCE ANALYSIS

For ease of presentation, without loss of generality, we set $f(X_t, t) \equiv 0$, $\beta(t, X_t) \equiv 1$, $u(t) \equiv 1$, $\sigma(t) \equiv \sigma_0$ for some $\alpha, \sigma_0 > 0$, and $\lambda(t) \equiv 0$. By this, the DiffFlow simplifies to the following SDE:

$$dX_t = \left[ \nabla \log \frac{q(X_t; \sigma_0)}{p_t(X_t)} + \frac{g^2(t)}{2} \nabla \log p_t(X_t) \right] dt + g(t) dW_t.$$

**Remark 1.** *The current simplified DiffFlow is general enough to incorporate the full dynamics of the original DiffFlow. By rescaling the particles and gradients, we can recover general $f(X_t, t)$, $\beta(t)$ and $u(t)$, as in DDIM (Song et al., 2020). The general forms of $\lambda(t)$ and $\sigma(t)$ can be recovered by modifying the noise scheduler from constant values to the desired annealing scheduler. The convergence analysis for annealing noise scheduler could be extended based on current analysis some techniques in Tang & Zhou (2021)*

To study the convergence of DiffFlow dynamics, we need to identify its variational formulation, i.e., a functional that DiffFlow minimizes. Finding such a functional is not difficult, as shown in Appendix E, where it is revealed that the functional is exactly the KL divergence between the generated and target distributions.

The main tool for proving the asymptotic convergence of DiffFlow is the Gaussian Poincaré inequality from Ledoux (2006):

**Definition 1** (Gaussian Poincaré Inequality). *Suppose $f : \mathbb{R}^k \to \mathbb{R}$ is a smooth function, and $X$ follows a multivariate Gaussian distribution $X \sim \mathcal{N}(0, \sigma^2 I)$, where $I \in \mathbb{R}^{k \times k}$. Then,*

$$Var[f(X)] \leq \sigma^2 \mathbb{E} \left[ \|\nabla f(X)\|_2^2 \right], \tag{13}$$

holds. We also show that under Gaussian smoothing, the log-density exhibits quadratic growth.

**Lemma 1** (Bound on the Log-Density of Smoothed Distributions, Proved in Appendix F). *Consider a probability distribution $q(x)$, and let $q(x; \sigma)$ be the distribution of $x + \epsilon$ where $x \sim q(x)$ and $\epsilon \sim \mathcal{N}(0, \sigma^2 I)$. Then, there exist constants $A_\sigma, B_\sigma > 0$, and $C_\sigma$ such that for any $0 < \gamma < 1$:*

$$|\log q(x; \sigma)| \leq A_\sigma \|x\|_2^2 + B_\sigma \|x\|_2 + C_\sigma, \tag{14}$$

*where $C_q(\gamma) := \inf \left\{ s : \int_{B(s)} q(u) du \geq \gamma \right\}$, $A_\sigma = \frac{1}{2\sigma^2}$, $B_\sigma = \frac{C_q(\gamma)}{\sigma^2}$, and $C_\sigma = \max \left\{ \frac{C_q^2(\gamma)}{2\sigma^2} - \log \left( \gamma \frac{1}{(2\pi)^{k/2} \sigma^k} \right), \log \left( \frac{1}{(2\pi)^{k/2} \sigma^k} \right) \right\}$, where $B(s) = \{x \in \mathbb{R}^k : \|x\|_2 \leq s\}$.*

**Remark 2.** *In the above lemma, for the measure $q$, we define the quantity $C_q(\gamma) := \inf \left\{ s : \int_{B(s)} q(u) du \geq \gamma \right\}$. This quantity $C_q(\gamma)$ represents the smallest ball centered at the origin in $\mathbb{R}^k$ that captures at least $\gamma$ mass of the probability measure $q$.*

Now, we are ready to prove the following asymptotic convergence theorem.

**Theorem 1** (Proved in Appendix G). *Consider a stochastic process $\{X_t\}_{t \geq 0}$ with dynamics determined by:*

$$dX_t = \left[ \nabla \log \frac{q(X_t; \sigma_0)}{p_t(X_t)} + \frac{g^2(t)}{2} \nabla \log p_t(X_t) \right] dt + g(t) dW_t, \tag{15}$$

*where $X_0 \sim \pi(x)$ and $\sigma_0, \lambda_0 > 0$. Then, the marginal distribution $X_t \sim p_t(x)$ converges almost everywhere to the target distribution $q(x; \sigma_0)$, i.e.:*

$$\lim_{t \to \infty} p_t(x) = q(x; \sigma_0) \quad a.e. \tag{16}$$

## 5    RELATED WORK

Our work is inspired by a recent series of studies that relate the training dynamics of GANs and SDMs to the particle evolution of ordinary or stochastic differential equations (Gao et al., 2019; Song & Ermon, 2019; Song et al., 2021b; 2020; Karras et al., 2022; Yi et al., 2023) and the references therein.

The training dynamics of vanilla GANs (Goodfellow et al., 2020) are highly unstable due to the minimax objective. Several subsequent works attempt to improve the stability of GAN training by considering particle gradient flow, which avoids the minimax framework (Nowozin et al., 2016; Arjovsky et al., 2017; Gulrajani et al., 2017; Gao et al., 2019; Yi et al., 2023). The key idea is that the evolution of particles can be driven by the gradient flow of a distance measure between probability distributions, such as KL divergence or Wasserstein distance. Consequently, the evolution dynamics can be described by an ODE, and the driven term of the ODE is determined by the functional gradient of the distance measure. For instance, in the case of KL divergence, the functional gradient corresponds to the gradient field of the logistic classifier (Gao et al., 2019; Yi et al., 2023), which plays the role of the discriminator in vanilla GANs.

The early development of diffusion models primarily focused on learning a series of Markov transition operators that maximize the ELBO (Sohl-Dickstein et al., 2015; Ho et al., 2020). In parallel, Song & Ermon (2019; 2020) proposed a set of score-based generative models based on multiple levels of denoising score matching. In 2020, Song et al. (2021b) demonstrated that diffusion models are essentially score-based generative models with score matching on multiple noise levels of corrupted target distributions. Furthermore, Song et al. (2021b) showed that the sampling dynamics of diffusion models can be modeled as stochastic differential equations with the scores of the noise-corrupted target serving as the drift term. Since then, diffusion models have been referred to as score-based diffusion models.

Although the training dynamics of both GANs and score-based diffusion models can be modeled by particle algorithms, whose dynamics are described by respective differential equations, there is currently no unified differential equation that can capture the dynamics of both. Our main contribution is to propose an SDE that enables the construction of a continuous spectrum that unifies GANs and diffusion models.

The contemporaneous work of Franceschi et al. (2023) proposed the Generative Particle Model (GPM) that unifies GANs and diffusion models. Our work shares the same motivation and idea with Franceschi et al. (2023) in the sense that both GANs and diffusion models can be viewed as a particle optimization algorithm. Our work differs from Franceschi et al. (2023) mainly in two aspects: i) we propose an explicit unified SDE for both GANs and diffusion models while there is no explicit unified SDE formulation for GPM. Under the GPM framework, GANs and diffusion models has different specific formulations of SDE or ODE. ii) the GPM paper provides much more empirical analysis and focus less on theory. Our work can be seen as a good complementary to the contemporaneous GPM paper.

## 6    CONCLUSION

We propose a unified SDE framework called "DiffFlow" that integrates the particle dynamics of the Langevin algorithm, GANs, diffusion SDEs, and diffusion ODEs. Our framework extends beyond SDMs and GANs, providing a continuous spectrum and enabling the development of new generative algorithms such as diffusion-GANs and SLCD. We conduct convergence analysis of DiffFlow and demonstrate that it allows for maximum likelihood estimation in both GANs and SDMs within the SDE framework.

However, our current framework only encompasses derivations of the density ratio-based GANs such as KL-GANs and vanilla GANs, and not all GANs (like IPM-based GANs, for example). It remains an open problem on how to incorporate the IPM-based GANs in to a unified framework.

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

## A   DERIVATION OF VP SDE

We aim to express the score of the forward process $\nabla \log p_t(X_t)$ in the VP SDE using the score of the noise-corrupted target distribution. We present the following proposition, which can be derived through a simple application of stochastic calculus, similar to Song et al. (2021b).

**Proposition 2.** *Assuming $X_0 \sim q(x)$ and the forward process $\{X_t\}_{t \in [0,T]}$ in the VP SDE is given by*

$$dX_t = -\frac{1}{2}\beta(t)X_t dt + \sqrt{\beta(t)}dW_t . \tag{17}$$

*Then the score of $p_t(x)$ (note: $X_t \sim p_t(x)$) can be represented as*

$$\nabla \log p_t(x) = \nabla \log q \left( \exp \left( \frac{1}{2}\int_0^t \beta(s)ds \right) x; 1 - \exp \left( -\int_0^t \beta(s)ds \right) \right) . \tag{18}$$

*Now, we can derive the denoising process of the VP SDE as follows:*

$$dX_t = \left[ \beta(T-t)\nabla \log q \left( \exp \left( \frac{1}{2}\int_0^{T-t} \beta(s)ds \right) X_t; 1 - \exp \left( -\int_0^{T-t} \beta(s)ds \right) \right) \right.$$
$$\left. + \frac{1}{2}\beta(T-t)X_t \right] dt + \sqrt{\beta(T-t)}dW_t.$$

*Proof.* By ito lemma, we have

$$d \left[ \exp \left( \frac{1}{2}\int_0^t \beta(s)ds \right) X_t \right] = \exp \left( \frac{1}{2}\int_0^t \beta(s)ds \right) dX_t + \frac{1}{2}\beta(t)\exp \left( \frac{1}{2}\int_0^t \beta(s)ds \right) X_t dt .$$

Combining with the forward VP SDE, we have

$$d \left[ \exp \left( \frac{1}{2}\int_0^t \beta(s)ds \right) X_t \right] = \exp \left( \frac{1}{2}\int_0^t \beta(s)ds \right) \sqrt{\beta(t)}dW_t . \tag{19}$$

Hence,

$$X_t = \exp \left( -\frac{1}{2}\int_0^t \beta(s)ds \right) \left[ X_0 + \int_0^t \exp \left( \frac{1}{2}\int_0^u \beta(s)ds \right) \sqrt{\beta(u)}dW_u \right] . \tag{20}$$

By the ito isometry and martingale property of brownian motions, we have

$$\mathbb{E}\left[ \int_0^t \exp \left( \frac{1}{2}\int_0^u \beta(s)ds \right) \sqrt{\beta(u)}dW_u \right] = 0 \tag{21}$$

and

$$\mathbb{E}\left[ \int_0^t \exp \left( \frac{1}{2}\int_0^u \beta(s)ds \right) \sqrt{\beta(u)}dW_u \right]^2$$
$$= \int_0^t \left[ \exp \left( \frac{1}{2}\int_0^u \beta(s)ds \right) \sqrt{\beta(u)} \right]^2 du$$
$$= \int_0^t \exp \left( \int_0^u \beta(s)ds \right) \beta(u)du$$
$$= \int_0^t \exp \left( \int_0^u \beta(s)ds \right) d\left[ \int_0^u \beta(s)du \right]$$
$$= \exp \left( \int_0^t \beta(s)ds \right) - 1 . \tag{22}$$

Hence

$$\exp \left( -\frac{1}{2}\int_0^t \beta(s)ds \right) \int_0^t \exp \left( \frac{1}{2}\int_0^u \beta(s)ds \right) \sqrt{\beta(u)}dW_u \sim \mathcal{N}\left( 0, I - \exp \left( -\int_0^t \beta(s)ds \right) I \right) . \tag{23}$$

Then we have

$$p_t(x) = \exp\left(\frac{1}{2}\int_0^t \beta(s)ds\right) q\left(\exp\left(\frac{1}{2}\int_0^t \beta(s)ds\right)x\right) \circledast \mathcal{N}\left(0, I - \exp\left(-\int_0^t \beta(s)ds\right)I\right)$$

(24)

where the proof ends by taking logarithm and applying the divergence operator $\nabla$ on both sides. $\square$

## B  DIFFODE'S EQUIVALENCE TO GANS

DiffODE naturally yield the following algorithm and we can show this algorithm is equivalent to GAN by setting the discriminator loss to logistic loss: $d_{\theta_D}(x) = \frac{1}{1+e^{-D_{\theta_D}(x)}}$.

---

**Algorithm 1** DiffFlow-GANs

---

**INPUT: target data from** $q(x;\sigma_0)$**:** $x_1^*, \ldots, x_n^* \in \mathbb{R}^k$**; noisy samples** $x_0^0, \ldots, x_n^0 \in \mathbb{R}^k$ **generated from noisy distribution** $\pi(x)$**; meta-parameter:** $T$**; neural network classifier:** $D_{\theta_D}(x)$**; generator:** $G_{\theta_G}(x)$**.**

> **for** $t = 1, \ldots, T$ **do**
>
> > Let $\theta_D^{t-1} = \arg\min_{\theta_D}\left[\frac{1}{n}\sum_{i=1}^n \log\left(1 + e^{-D_{\theta_D}(x_i^*)}\right) + \frac{1}{n}\sum_{i=1}^n \log\left(1 + e^{D_{\theta_D}(x_i^{t-1})}\right)\right]$
> >
> > Sample $z_1, \ldots, z_n$ from noisy prior $\pi(x)$.
> >
> > Update the generator by descending the following loss:
> >
> > $$\frac{1}{2n}\sum_{i=1}^n \|G_{\theta_G}(z_i) - (G_{\theta_G^{t-1}}(z_i) + \eta_t\beta(z_i, t)\nabla D_{\theta_D^{t-1}}(G_{\theta_G^{t-1}}(z_i)))\|_2^2 .$$
> >
> > Update the sampled particles: $x_i^t = x_i^{t-1} + \eta_t\beta(x_i^{t-1}, t)\nabla D_{\theta_D^{t-1}}(x_i^{t-1})$ for $i = 1, \ldots, n$
>
> **end for**
> **return** $G_{\theta_G^T}(x)$ and particles $\{x_i^T\}_{i=1}^n$ .

---

For the vanilla GANs (Goodfellow et al., 2020), the update of discriminator is the same as DiffODE-GANs. Since from $d_{\theta_D}(x) = \frac{1}{1+e^{-D_{\theta_D}(x)}}$, we have

$$\mathbb{E}_{x\sim q(x;\sigma_0)}\log\left(1 + e^{-D_{\theta_D}(x)}\right) + \mathbb{E}_{x\sim p_t(x)}\log\left(1 + e^{D_{\theta_D}(x)}\right)$$

$$= \mathbb{E}_{x\sim q(x;\sigma_0)}\log\left(1 + e^{-\log\frac{d_{\theta_D}(x)}{1-d_{\theta_D}(x)}}\right) + \mathbb{E}_{x\sim p_t(x)}\log\left(1 + e^{\log\frac{d_{\theta_D}(x)}{1-d_{\theta_D}(x)}}\right)$$

$$= -\mathbb{E}_{x\sim q(x;\sigma_0)}\log\left(d_{\theta_D}(x)\right) - \mathbb{E}_{x\sim p_t(x)}\log\left(1 - d_{\theta_D}(x)\right) .$$

(25)

It remains to show that the update of the generator is also equivalent. The gradient of the generator is

$$\begin{aligned}
&\nabla_{\theta_G}\log(1 - d_{\theta_D}(G_{\theta_G}(z)))\\
&= -\nabla_{\theta_G}\log(1 + e^{D_{\theta_D}(G_{\theta_G}(z))})\\
&= -\frac{1}{1 + e^{-D_{\theta_D}(G_{\theta_G}(z))}}\nabla_{\theta_G}D_{\theta_D}(G_{\theta_G}(z))\\
&= -d_{\theta_D}(G_{\theta_G}(z))\nabla_{\theta_G}D_{\theta_D}(G_{\theta_G}(z))\\
&= -d_{\theta_D}(G_{\theta_G}(z))\nabla D_{\theta_D}(G_{\theta_G}(z)) \cdot \nabla_{\theta_G}G_{\theta_G}(z) .
\end{aligned}$$

(26)

Hence, at the time step $t - 1$, we obtain the discriminator with parameter $\theta_D^{t-1}$ and update generator by the following equation

$$\theta_G^t = \theta_G^{t-1} + \lambda_t\left[\frac{1}{n}\sum_{i=1}^n d_{\theta_D^{t-1}}(G_{\theta_G^{t-1}}(z_i))\nabla D_{\theta_D^{t-1}}(G_{\theta_G^t}(z_i)) \cdot \nabla_{\theta_G}G_{\theta_G^{t-1}}(z)\right]$$

(27)

where $z_i \sim \mathcal{N}(0, I)$ and $\lambda_t$ is the learning rate for mini-batch SGD at time $t$.

If we instead run a gradient descent step on the MSE loss of the generator in the DiffFlow-GAN, we obtain

$$\theta_G^t = \theta_G^{t-1} + \lambda_t \left[ \frac{1}{n} \sum_{i=1}^n \eta_t \beta(z_i, t) \nabla D_{\theta_D^t}(G_{\theta_G^t}(z_i)) \cdot \nabla_{\theta_G} G_{\theta_G^t}(z_i) \right] . \tag{28}$$

Then the equivalence can be shown by setting $\eta_t \beta(z_i, t) = d_{\theta_D^{t-1}}(G_{\theta_G^{t-1}}(z_i))$ .

In practice, the vanilla GAN faces the problem of gradient vanishing on the generator update. A common trick applied is to use the "non-saturating loss", i.e., the generator update is by instead minimizing $-\mathbb{E}_{z \sim \pi(z)}[\log(d_{\theta_D}(G_{\theta_G}(z)))]$. Hence, the gradient of the generator is

$$\begin{aligned}
&-\nabla_{\theta_G} \log(d_{\theta_D}(G_{\theta_G}(z))) \\
&= \nabla_{\theta_G} \log(1 + e^{-D_{\theta_D}(G_{\theta_G}(z))}) \\
&= -\frac{e^{-D_{\theta_D}(G_{\theta_G}(z))}}{1 + e^{-D_{\theta_D}(G_{\theta_G}(z))}} \nabla_{\theta_G} D_{\theta_D}(G_{\theta_G}(z)) \\
&= -(1 - d_{\theta_D}(G_{\theta_G}(z))) \nabla_{\theta_G} D_{\theta_D}(G_{\theta_G}(z)) \\
&= -(1 - d_{\theta_D}(G_{\theta_G}(z))) \nabla D_{\theta_D}(G_{\theta_G}(z)) \cdot \nabla_{\theta_G} G_{\theta_G}(z) .
\end{aligned} \tag{29}$$

Similarly, with the discriminator parameter $\theta_D^{t-1}$, we can update generator by the following equation

$$\theta_G^t = \theta_G^{t-1} + \lambda_t \left[ \frac{1}{n} \sum_{i=1}^n (1 - d_{\theta_D^{t-1}}(G_{\theta_G^{t-1}}(z_i))) \nabla D_{\theta_D^{t-1}}(G_{\theta_G^t}(z_i)) \cdot \nabla_{\theta_G} G_{\theta_G^{t-1}}(z_i) \right] \tag{30}$$

Then the equivalence can be shown by setting $\eta_t \beta(z_i, t) = 1 - d_{\theta_D^{t-1}}(G_{\theta_G^{t-1}}(z_i))$ .

**Remark 3.** *The DiffFlow-GANs formulation provides a more intuitive explanation on why non-saturating loss can avoid vanishing gradients for a poor-trained generator: if at time $t - 1$, we have a poor generator $G_{\theta_G^{t-1}}(z)$, generating poor samples that are far from the real data; then $d_{\theta_D^{t-1}}(G_{\theta_G^{t-1}}(z_i))$ would close to 0, which would lead to zero particle update gradient for original GANs while the "non-saturating" loss can avoid this problem.*

*In practice, one can avoid the gradient vanishing for DiffFLow-GANs by two methods: either by setting $\beta(z_i, t) \equiv 1$ to maintain the gradient for particle updates; or proposing a noising annealing strategy for the discriminator: during the early stage of training, the discriminator is weakened by classifying a noise-corrupted target distribution $q(x; \sigma(t))$ from fake data $p_t(x)$. The weakening discriminator trick has been adopted in many real deployed GAN models, and it has been shown helpful during the early stage of GAN training (Salimans et al., 2016). The noise annealing on discriminator shares some spirits with SDMs. We will discuss this point in details in the next Appendix C.*

## C  THREE IMPROVEMENTS ON VANILLA GANS

From previous analysis, we show that the DiffFlow framework provides a novel view on GANs and has potential for several improvements on the vanilla GANs algorithm. The generation dynamics of vanilla GANs are coarse approximation of DiffODE: the one-step gradient of the generator is determined by the particle movements driven by DiffODE, and the driven force is exactly the gradient field of the logistic classifier between the real data and fake data (i.e., discriminator). Furthermore, for the vanilla GAN, the particle gradient is scaled by the probability of th e particle being real data, which would be near zero at the early stage of training — this is exactly the source of gradient vanishing. From this perspective, we can obtain the following improvements: simplify $\beta(t, X_t) \equiv \beta(t)$, i.e., eliminating the dependence of the particle movement on the scaling factor that determined by the probability of the particle being real. This would alleviate the gradient vanishing at the early stage of training.

Furthermore, since the vanilla GAN only approximates the particle movements by one-step gradients of the least square regression, the is too coarse to simulate the real dynamics of DiffODE. Indeed,

one could directly composite a generator by the gradient fields of discriminators at each time step $t$ and this generator could directly simulate the original particle movements in the DiffODE. The idea can be implemented by borrowing ideas from diffusion models: we adopt a time-indexed neural network discriminator $d_{\theta_t}(x, t)$ that is trained by classifying real and fake data at time $t$.

Lastly, since at the early stage of training, the generator could face too much pressure with a "smart" discriminator, the transition and training dynamics between noise to data could be sharp and unstable during the early stage of training. To achieve a smooth the transition between noise to data, we borrow again ideas from diffusion models: we adopt a the noise annealing strategy $\sigma(t)$ that weakening the discriminator. At time $t$, the discriminator $d_{\theta_t}(x, t)$ learns to classify between noise-corrupted real data $q(x; \sigma(t))$ and fake data $p_t(x)$ where the corruption $\sigma(t)$ is continuously decreasing as time index increasing with $\sigma(0) = \sigma_{max}$ and $\sigma(T) = \sigma_{min}$. This idea is analogous to diffusion models such as NCSN (Song & Ermon, 2019) and the only difference is that the diffusion models learn the score of noise corrupted target distribution $q(x; \sigma(t))$ instead of a classifier at the time index $t$.

With above three improvements inspired from the perspective of ODE approximations and diffusion models, we propose an improved GAN algorithm. The training and sampling procedure is described as follows.

---

**Algorithm 2** Improved-DiffFlow-GANs-Training

**INPUT: target data from** $q(x)$**:** $x_1^*, \ldots, x_n^* \in \mathbb{R}^k$**; noise annealing strategy** $\{\sigma_i\}_{i=1}^T$**; noisy samples** $x_0^0, \ldots, x_n^0 \in \mathbb{R}^k$ **generated from noisy distribution** $\pi(x)$**; meta-parameter:** $T$**; time-indexed neural network classifier:** $D_{\theta_t}(x, t)$**.**

   **for** $t = 1, \ldots, T$ **do**

      Sample $z_1, \ldots, z_n$ from $\mathcal{N}(0, \sigma_i^2 I)$.

      Let $\theta_t^* = \arg\min_{\theta_D} \left[ \frac{1}{n} \sum_{i=1}^n \log \left( 1 + e^{-D_{\theta_t}(x_i^* + z_i, t)} \right) + \frac{1}{n} \sum_{i=1}^n \log \left( 1 + e^{D_{\theta_t}(x_i^{t-1}, t)} \right) \right]$

      Update the sampled particles: $x_i^t = x_i^{t-1} + \eta_t \beta(t) \nabla D_{\theta_t^*}(x_i^{t-1}, t)$ for $i = 1, \ldots, n$

   **end for**

   **return** Particles $\{x_i^T\}_{i=1}^n$ and $\{\theta_t^*\}_{t=1}^T$.

---

**Algorithm 3** Improved-DiffFlow-GANs-Sampling

**INPUT: Noisy distribution** $\pi(x)$**; time-indexed discriminator:** $D_{\theta_t^*}(x, t)$ **for** $t = 1, \ldots, T$**.**

   Sample $X_0 \sim \pi(x)$

   **for** $t = 1, \ldots, T$ **do**

      Update the sampled particles: $X_t = X_{t-1} + \eta_t \beta(t) \nabla D_{\theta_t^*}(x_{t-1}, t-1)$

   **end for**

   **return** $X_T$.

---

Although we adopt the method of training discriminators $\theta_t^*$ independently across time steps in the algorithm's pseudocode, since it avoids the slow convergence that is partly due to conflicting optimization directions between different time steps (Hang et al., 2023). It worths mentioning that our framework offers much more flexibility on designing the time-indexed discriminator: we can either share a universal $\theta$ across all time $t$ as done in diffusion models, or train discriminators $\theta_t^*$ independently for each $t$. It remains an open problem on which method is better for such generative models.

Notice that in a recent work of MonoFlow (Yi et al. (2023)), the authors also discussed the problem of gradient vanishing in vanilla GANs arises from too small rescaled vector fields of discriminator-guided particle dynamics and it fixes the gradient vanishing problem by simply adding a constant in the generator output combined with a monotonic function (section 5.2 of Yi et al. (2023)).

It's also worth mentioning that the noise-corrupting strategies for Vanilla GANs mentioned in previous sections are strongly linked to preexisting works leveraging noise to regularize the discriminator,

e.g. instance noise (Sønderby et al. (2016)) and diffusion GANs (Wang et al. (2022)). The smoothed KL divergence in the DiffFlow framework is a generalization of the instance noise training objective of Sønderby et al. (2016)) with varying levels of noise.

The diffusion GAN algorithm from (Wang et al. (2022)) is quite different from the diffusion-GAN in our framework: in each discriminator-generator optimization cycle, (Wang et al. (2022)) propose to train and combine a series of discriminators that are trained by corrupting instance with different levels of noise. Since the noise schedule is aligned with diffusion models, the author name it diffusion GAN. The diffusion GAN in (Wang et al. (2022)) is essentially a GAN-like algorithm. While in our framework of diffusion GAN, we should jointly train a score network and GAN-flavoured generator and the generation process is guided by both networks. Hence, it is a real unified combination of GANs and diffusion models.

There are some more works on designing hybrid algorithms of GANs and diffusion models. For example, Kim et al. (2022) use discriminators with diffusion models as a refining process. Kim et al. (2022) has fundamental difference from our current framework: given a **fixed** pretrained score network $\nabla \log p_{model}(x_t, t)$ that may deviates from real data distribution, the authors train an additional time-dependent discriminator $d(x_t, t)$ that discriminates between the real data and generated data at time step $t$ and correct the score network by

$$\nabla \log p_{new}(x_t, t) = \nabla \log p_{model}(x_t, t) + \nabla \log \frac{d(x_t, t)}{1 - d(x_t, t)} .$$

In our framework, the score network and the discriminator network should be trained **jointly**. Furthermore, the noise in the discriminator should align with the diffusion process. Contrastly in Kim et al. (2022), there are no noise added on the target distribution $q(x)$ during the training of discriminator.

## D    PROOF OF PROPOSITION 1

*Proof.* By the Kolmogorov Forward Equation (Øksendal & Øksendal, 2003), the marginal distribution $p_t(x)$ follows the following PDE:

$$\frac{\partial p_t(x)}{\partial t}$$
$$= -\nabla \cdot \left[ p_t(x) \left( f(x, t) + \beta(t) \nabla \log \frac{q(u(t)x; \sigma(t))}{p_t(x)} + \frac{g^2(t)}{2} \nabla \log p_t(x) \right) \right] + \frac{g^2(t) - \lambda^2(t)}{2} \nabla \cdot \nabla p_t(x)$$
$$= -\nabla \cdot \left[ p_t(x) \left( f(x, t) + \beta(t) \nabla \log \frac{q(u(t)x; \sigma(t))}{p_t(x)} \right) \right] - \nabla \cdot \left[ p_t(x) \frac{g^2(t)}{2} \nabla \log p_t(x) \right] + \frac{g^2(t) - \lambda^2(t)}{2} \nabla \cdot \nabla p_t(x)$$
$$= -\nabla \cdot \left[ p_t(x) \left( f(x, t) + \beta(t) \nabla \log \frac{q(u(t)x; \sigma(t))}{p_t(x)} \right) \right] - \frac{\lambda^2(t)}{2} \nabla \cdot \nabla p_t(x) . \tag{31}$$

Hence, the marginal distribution $p_t(x)$ is independent of $g(\cdot)$. $\qquad\square$

## E    THE VARIATIONAL FORMULATION OF DIFFFLOW

**Lemma 2** (The Variational Formulation of DiffFlow). *Given stochastic process $\{X_t\}_{t\geq 0}$ and its dynamics determined by*

$$dX_t = \left[ \nabla \log \frac{q(X_t; \sigma_0)}{p_t(X_t)} + \frac{g^2(t)}{2} \nabla \log p_t(X_t) \right] dt + g(t) dW_t \tag{32}$$

*with $X_0 \sim \pi(x)$ and $\sigma_0, \lambda_0 > 0$. Then the marginal distribution $p_t(x)$ of $X_t$ minimizes the following functional*

$$L(p) = KL(p \| q(x; \sigma_0)) := \int_{\mathbb{R}^k} p(x) \log \frac{p(x)}{q(x; \sigma_0)} dx . \tag{33}$$

*Furthermore,*

$$\frac{\partial L(p_t)}{\partial t} = -\int_{\mathbb{R}^k} p_t(x) \left\| \nabla \log \frac{p_t(x)}{q(x; \sigma_0)} \right\|_2^2 dx . \tag{34}$$

*Proof.* By the Kolmogorov Forward Equation, the marginal distribution $p_t(x)$ follows the following PDE:

$$\frac{\partial p_t(x)}{\partial t}$$

$$= -\nabla \cdot \left[ p_t(x) \left( \nabla \log \frac{q(x; \sigma_0)}{p_t(x)} + \frac{g^2(t)}{2} \nabla \log p_t(x) \right) \right] + \frac{g^2(t)}{2} \nabla \cdot \nabla p_t(x)$$

$$= -\nabla \cdot \left[ p_t(x) \left( \nabla \log \frac{q(x; \sigma_0)}{p_t(x)} + \frac{g^2(t)}{2} \nabla \log p_t(x) \right) \right] + \frac{g^2(t)}{2} \nabla \cdot \left[ p_t(x) \nabla \log p_t(x) \right]$$

$$= -\nabla \cdot \left[ p_t(x) \left( \nabla \log \frac{q(x; \sigma_0)}{p_t(x)} + \frac{g^2(t)}{2} \nabla \log p_t(x) - \frac{g^2(t)}{2} \nabla \log p_t(x) \right) \right]$$

$$= -\nabla \cdot \left[ p_t(x) \left( \nabla \log \frac{q(x; \sigma_0)}{p_t(x)} \right) \right] . \tag{35}$$

Then, we have

$$\frac{\partial L(p_t)}{\partial t}$$

$$= \int_{\mathbb{R}^k} \left[ \log \frac{p_t(x)}{q(x; \sigma_0)} + 1 \right] \frac{\partial p_t(x)}{\partial t} dx$$

$$= \int_{\mathbb{R}^k} \log \frac{p_t(x)}{q(x; \sigma_0)} \frac{\partial p_t(x)}{\partial t} dx$$

$$= -\int_{\mathbb{R}^k} \log \frac{p_t(x)}{q(x; \sigma_0)} \nabla \cdot \left[ p_t(x) \left( \nabla \log \frac{q(x; \sigma_0)}{p_t(x)} \right) \right] dx$$

$$= \int_{\mathbb{R}^k} \log \frac{p_t(x)}{q(x; \sigma_0)} \nabla \cdot \left[ p_t(x) \left( \nabla \log \frac{p_t(x)}{q(x; \sigma_0)} \right) \right] dx \tag{36}$$

Through integral by parts, we have

$$\frac{\partial L(p_t)}{\partial t}$$

$$= -\int_{\mathbb{R}^k} p_t(x) \left( \nabla \log \frac{p_t(x)}{q(x; \sigma_0)} \right) \cdot \left( \nabla \log \frac{p_t(x)}{q(x; \sigma_0)} \right) dx$$

$$= -\int_{\mathbb{R}^k} p_t(x) \left\| \nabla \log \frac{p_t(x)}{q(x; \sigma_0)} \right\|_2^2 dx . \tag{37}$$

Hence, the KL divergence $L(p_t)$ is decreasing along the marginal distribution path $\{p_t(x)\}_{t \geq 0}$ determined by DiffFlow. $\qquad \square$

## F  PROOF OF LEMMA 1

*Proof.* Let $G_\sigma(x)$ be the probability density function of $N(0, \sigma^2 I)$, then the resulting smoothed distribution $q(x; \sigma)$ is

$$q(x; \sigma) = \int_{\mathbb{R}^d} q(u) G_\sigma(x - u) du . \tag{38}$$

Let $B(r) = \{u : \|u\|_2 \leq r\}$, then

$$q(x; \sigma) = \int_{B(r)} q(u) G_\sigma(x - u) du + \int_{\mathbb{R}^k \setminus B(r)} q(u) G_\sigma(x - u) du$$

$$\geq \int_{B(r)} q(u) G_\sigma(x - u) du$$

$$\geq \int_{B(r)} q(u) G_\sigma \left( x + r \frac{x}{\|x\|_2} \right) du$$

$$= G_\sigma \left( x + r \frac{x}{\|x\|_2} \right) \int_{B(r)} q(u) du . \tag{39}$$

Fix some small constant $0 < \gamma < 1$, if we choose $r = C_p(\gamma) := \inf \left\{ s : \int_{B(s)} p(u)du \geq \gamma \right\}$. This implies

$$
\begin{aligned}
q(x; \sigma) &\geq \gamma G_\sigma \left( x + C_p(\gamma) \frac{x}{\|x\|_2} \right) \\
&= \gamma \frac{1}{(2\pi)^{k/2}\sigma^k} \exp \left( \frac{-\left\| x + C_p(\gamma) \frac{x}{\|x\|_2} \right\|_2^2}{2\sigma^2} \right),
\end{aligned}
\tag{40}
$$

Taking logarithm on both sides of (40), we obtain

$$
\begin{aligned}
\log q(x; \sigma) &\geq \log \left( \gamma \frac{1}{(2\pi)^{k/2}\sigma^k} \right) - \frac{\left\| x + C_p(\gamma) \frac{x}{\|x\|_2} \right\|_2^2}{2\sigma^2} \\
&= -\frac{1}{2\sigma^2}\|x\|_2^2 - \frac{C_p(\gamma)}{\sigma^2}\|x\|_2 - \frac{C_p^2(\gamma)}{2\sigma^2} + \log \left( \gamma \frac{1}{(2\pi)^{k/2}\sigma^k} \right).
\end{aligned}
\tag{41}
$$

We also have

$$
\begin{aligned}
q(x; \sigma) &= \int_{\mathbb{R}^k} q(u)G_\sigma(x - u)du \\
&\leq \int_{\mathbb{R}^k} q(u)G_\sigma(0)du \\
&= G_\sigma(0) \\
&= \frac{1}{(2\pi)^{k/2}\sigma^k}.
\end{aligned}
\tag{42}
$$

Therefore,

$$
\log q(x; \sigma) \leq \log \left( \frac{1}{(2\pi)^{k/2}\sigma^k} \right).
\tag{43}
$$

Let $A_\sigma = \frac{1}{2\sigma^2}, B_\sigma = \frac{C_p(\gamma)}{\sigma^2}$ and $C_\sigma = \max \left\{ \frac{C_p^2(\gamma)}{2\sigma^2} - \log \left( \gamma \frac{1}{(2\pi)^{k/2}\sigma^k} \right), \log \left( \frac{1}{(2\pi)^{k/2}\sigma^k} \right) \right\}$, then

$$
|\log q(x; \sigma)| \leq A_\sigma \|x\|_2^2 + B_\sigma \|x\|_2 + C_\sigma.
\tag{44}
$$

$\square$

## G  PROOF OF THEOREM 1

*Proof.* By Lemma 2,

$$
\frac{\partial L(p_t)}{\partial t} = -\int_{\mathbb{R}^k} p_t(x) \left\| \nabla \log \frac{p_t(x)}{q(x; \sigma_0)} \right\|_2^2 dx.
\tag{45}
$$

Hence, by the nonnegativity of KL divergence,

$$
\lim_{t \to \infty} \int_{\mathbb{R}^k} p_t(x) \left\| \nabla \log \frac{p_t(x)}{q(x; \sigma_0)} \right\|_2^2 dx = 0.
\tag{46}
$$

Furthermore, since

$$
\left\| \nabla \sqrt{f(x)} \right\|_2^2 = \frac{\|\nabla f(x)\|_2^2}{4f(x)} = \frac{f(x)}{4} \|\nabla \log f(x)\|_2^2
\tag{47}
$$

we have

$$\int_{\mathbb{R}^k} p_t(x) \left\| \nabla \log \frac{p_t(x)}{q(x;\sigma_0)} \right\|_2^2 dx$$

$$= 4 \int_{\mathbb{R}^k} q(x;\sigma_0) \left\| \nabla \sqrt{\frac{p_t(x)}{q(x;\sigma_0)}} \right\|_2^2 dx$$

$$= 4 \int_{\mathbb{R}^k} \exp\left(\log q(x;\sigma_0)\right) \left\| \nabla \sqrt{\frac{p_t(x)}{q(x;\sigma_0)}} \right\|_2^2$$

$$\geq 4 \int_{\mathbb{R}^k} \exp\left(-A_{\sigma_0}\|x\|_2^2 - B_{\sigma_0}\|x\|_2 - C_{\sigma_0}\right) \left\| \nabla \sqrt{\frac{p_t(x)}{q(x;\sigma_0)}} \right\|_2^2 dx . \qquad (48)$$

Since $\|x\|_2 \leq \|x\|_2^2 + 1$, we have

$$\int_{\mathbb{R}^k} p_t(x) \left\| \nabla \log \frac{p_t(x)}{q(x;\sigma_0)} \right\|_2^2 dx$$

$$\geq 4 \int_{\mathbb{R}^k} \exp\left(-(A_{\sigma_0} + B_{\sigma_0})\|x\|_2^2 - C_{\sigma_0} - 1\right) \left\| \nabla \sqrt{\frac{p_t(x)}{q(x;\sigma_0)}} \right\|_2^2 dx$$

$$= 4 \exp\left(-C_{\sigma_0} - 1\right) \int_{\mathbb{R}^k} \exp\left(-\frac{\|x\|_2^2}{(A_{\sigma_0} + B_{\sigma_0})^{-1}}\right) \left\| \nabla \sqrt{\frac{p_t(x)}{q(x;\sigma_0)}} \right\|_2^2 dx$$

$$= 4 \left(\frac{\pi}{A_{\sigma_0} + B_{\sigma_0}}\right)^{k/2} \exp\left(-C_{\sigma_0} - 1\right) \int_{\mathbb{R}^k} \mathcal{N}\left(x; 0, \frac{1}{2(A_{\sigma_0} + B_{\sigma_0})}I\right) \left\| \nabla \sqrt{\frac{p_t(x)}{q(x;\sigma_0)}} \right\|_2^2 dx$$

$$= 4 \left(\frac{\pi}{A_{\sigma_0} + B_{\sigma_0}}\right)^{k/2} \exp\left(-C_{\sigma_0} - 1\right) \mathbb{E}_{x \sim \mathcal{N}\left(x; 0, \frac{1}{2(A_{\sigma_0}+B_{\sigma_0})}I\right)} \left(\left\| \nabla \sqrt{\frac{p_t(x)}{q(x;\sigma_0)}} \right\|_2^2\right)$$

$$\geq 8 (A_{\sigma_0} + B_{\sigma_0}) \left(\frac{\pi}{A_{\sigma_0} + B_{\sigma_0}}\right)^{k/2} \exp\left(-C_{\sigma_0} - 1\right) Var_{x \sim \mathcal{N}\left(x; 0, \frac{1}{2(A_{\sigma_0}+B_{\sigma_0})}I\right)} \left(\sqrt{\frac{p_t(x)}{q(x;\sigma_0)}}\right) .$$

where the last inequality is due to Gaussian Poincare inequality. Hence, we obtain

$$\lim_{t\to\infty} Var_{x \sim \mathcal{N}\left(x; 0, \frac{1}{2(A_{\sigma_0}+B_{\sigma_0})}I\right)} \left(\sqrt{\frac{p_t(x)}{q(x;\sigma_0)}}\right) = 0 . \qquad (49)$$

Furthermore, by previous analysis on the lower bound of $\exp\left(\log q(x;\sigma_0)\right)$, we have

$$\exp\left(\log q(x;\sigma_0)\right) \geq D_{\sigma_0} \mathcal{N}\left(x; 0, \frac{1}{2(A_{\sigma_0} + B_{\sigma_0})}I\right) . \qquad (50)$$

where $D_{\sigma_0} := 4 \left(\frac{\pi}{A_{\sigma_0} + B_{\sigma_0}}\right)^{k/2} \exp\left(-C_{\sigma_0} - 1\right)$.

Then

$$\mathbb{E}_{x \sim \mathcal{N}\left(x; 0, \frac{1}{2(A_{\sigma_0} + B_{\sigma_0})} I\right)} \left( \sqrt{\frac{p_t(x)}{q(x; \sigma_0)}} \right)$$

$$= \int_{\mathbb{R}^k} \mathcal{N}\left(x; 0, \frac{1}{2(A_{\sigma_0} + B_{\sigma_0})} I\right) \sqrt{\frac{p_t(x)}{q(x; \sigma_0)}} dx$$

$$\leq \frac{1}{\sqrt{D_{\sigma_0}}} \int_{\mathbb{R}^k} \sqrt{\mathcal{N}\left(x; 0, \frac{1}{2(A_{\sigma_0} + B_{\sigma_0})} I\right)} \sqrt{p_t(x)} dx$$

$$\leq \frac{1}{\sqrt{D_{\sigma_0}}} \sqrt{\int_{\mathbb{R}^k} \mathcal{N}\left(x; 0, \frac{1}{2(A_{\sigma_0} + B_{\sigma_0})} I\right) dx \int_{\mathbb{R}^k} p_t(x) dx}$$

$$= \frac{1}{\sqrt{D_{\sigma_0}}} . \tag{51}$$

Hence,

$$\lim_{t \to \infty} \sqrt{\frac{p_t(x)}{q(x; \sigma_0)}} = const. \leq \frac{1}{\sqrt{D_{\sigma_0}}} < \infty \tag{52}$$

This implies

$$\lim_{t \to \infty} p_t(x) = q(x; \sigma_0) \quad \text{a.e.} \tag{53}$$

$\square$

## H  NONASYMPTOTIC CONVERGENCE ANALYSIS VIA LOG-SOBOLEV INEQUALITY

In the previous section, we have proved the asymptotic optimality of DiffFlow. In order to obtain an explicit convergence rate to the target distribution, we need stronger functional inequalities, i.e., the log-Sobolev inequality (Ledoux, 2006; Ma et al., 2019).

**Definition 2** (Log-Sobolev Inequality). *For a smooth function $g : \mathbb{R}^k \to \mathbb{R}$, consider the Sobolev space defined by the weighted $L^2$ norm: $\|g\|_{L^2(q)} = \int_{\mathbb{R}^k} g(x)^2 q(x) dx$. We say $q(x)$ satisfies the log-Sobolev inequality with constant $\rho > 0$ if the following inequality holds for any $\int_{\mathbb{R}^k} g(x) q(x) = 1$,*

$$\int_{\mathbb{R}^k} g(x) \log g(x) \cdot q(x) dx \leq \frac{2}{\rho} \int_{\mathbb{R}^k} \left\| \nabla \sqrt{g(x)} \right\|_2^2 q(x) dx . \tag{54}$$

Then we can obtain a linear convergence of marginal distribution $p_t(x)$ to the smoothed target distribution $q(x; \sigma_0)$. The analysis is essentially the same as Langevin dynamics as in (Ma et al., 2019), since their marginal distribution shares the same Fokker-Planck equation.

**Theorem 2.** *Given stochastic process $\{X_t\}_{t \geq 0}$ and its dynamics determined by*

$$dX_t = \left[ \nabla \log \frac{q(X_t; \sigma_0)}{p_t(X_t)} + \frac{g^2(t)}{2} \nabla \log p_t(X_t) \right] dt + g(t) dW_t \tag{55}$$

*with $X_0 \sim \pi(x)$ and $\sigma_0, \lambda_0 > 0$. If the smoothed target distribution $q(x; \sigma_0)$ satisfies the log-Sobolev inequality with constant $\rho_q > 0$. Then the marginal distribution $X_t \sim p_t(x)$ converges linear to the target distribution $q(x; \sigma_0)$ in KL divergence, i.e.,*

$$KL(p_t(x) \| q(x; \sigma_0)) \leq \exp(-2\rho_q t) KL(\pi(x) \| q(x; \sigma_0)). \tag{56}$$

*Proof.* From Lemma 2, we obtain

$$\frac{\partial L(p_t)}{\partial t} = \frac{\partial}{\partial t} KL(p_t(x) \| q(x; \sigma_0)) = - \int_{\mathbb{R}^k} p_t(x) \left\| \nabla \log \frac{p_t(x)}{q(x; \sigma_0)} \right\|_2^2 dx . \tag{57}$$

Since $q(x; \sigma_0)$ satisfies the log-Sobolev inequality with constant $\rho_q > 0$, by letting the test function $g(x) = p_t(x)/q(x; \sigma_0)$, we obtain

$$KL(p_t(x)\|q(x; \sigma_0))$$

$$\leq \frac{2}{\rho_q} \int_{\mathbb{R}^k} \left\| \nabla \sqrt{\frac{p_t(x)}{q(x; \sigma_0)}} \right\|_2^2 q(x; \sigma_0) dx \ . \tag{58}$$

Since we already know from previous analysis that

$$\left\| \nabla \sqrt{f(x)} \right\|_2^2 = \frac{f(x)}{4} \left\| \nabla \log f(x) \right\|_2^2$$

we have

$$KL(p_t(x)\|q(x; \sigma_0))$$

$$\leq \frac{1}{2\rho_q} \int_{\mathbb{R}^k} p_t(x) \left\| \nabla \log \frac{p_t(x)}{q(x; \sigma_0)} \right\|_2^2 q(x) dx$$

$$= -\frac{1}{2\rho_q} \frac{\partial}{\partial t} KL(p_t(x)\|q(x; \sigma_0)) \ . \tag{59}$$

Hence by Grönwall's inequality,

$$KL(p_t(x)\|q(x; \sigma_0)) \leq \exp(-2\rho_q t) KL(\pi(x)\|q(x; \sigma_0)). \tag{60}$$

□

Different from asymptotic analysis in the previous subsection that holds for general $q(x)$, the convergence rate is obtained under the assumption that the smoothed target $q(x; \sigma_0)$ satisfies the log-Sobolev inequalities. This is rather a strong assumption, since we need control the curvature lower bound of smoothed energy function $U(x; q; \sigma_0)$ to satisfy the Lyapunov conditions for log-Sobolev inequality (Cattiaux et al., 2010). This condition holds for some simple distribution: if $q(x) \sim \mathcal{N}(\mu_q, \sigma_q^2 I)$, then the Hessian of its smoothed energy function $U(x; q; \sigma_0) = -\nabla^2 \log q(x; \sigma_0) = (\sigma_0^2 + \sigma_q^2)^{-1} I$, then by Bakery-Emery criteria (Bakry & Émery, 2006), we have $\rho_q = (\sigma_0^2 + \sigma_q^2)^{-1}$ . However, for a general target $q(x)$, obtaining the log-Sobolev inequality is relatively hard. If we still want to obtain an explicit convergence rate, we can seek for an explicit regularization $f(t, X_t)$ on the DiffFlow dynamics to restrict the path measure in a smaller subset and then the explicit convergence rate can be obtained by employing uniform log-Sobolev inequalities along the path measure (Guillin et al., 2022). The detailed derivation of such log-Sobolev inequalities under particular regularization $f(t, X_t)$ on the path measure is beyond the scope of this paper and we leave it as future work.

## I   MAXIMAL LIKELIHOOD INFERENCE

Recall that the dynamics of DiffFlow is described by the following SDE on the interval $t \in [0, T]$ with $X_0 \sim \pi(x)$ and $X_t \sim p_t(x)$:

$$dX_t = \left[ f(X_t, t) + \beta(t, X_t) \nabla \log \frac{q(u(t)X_t; \sigma(t))}{p_t(X_t)} + \frac{g^2(t)}{2} \nabla \log p_t(X_t) \right] dt + \sqrt{g^2(t) - \lambda^2(t)} dW_t \ . \tag{61}$$

At the end time step $T$, we obtain $p_T(x)$ and we hope this is a "good" approximation to the target distribution. Previous analysis shows that as $T$ to infinity, $p_T(x)$ would converge to the target distribution $q(x; \sigma_0)$ under simplified dynamics. However, the convergence rate is relatively hard to obtain for general target distribution with no isoperimetry property. In this section, we define the goodness of approximation of $p_T(x)$ from another perspective: maximizing the likelihood at the end of time.

There are already many existing works on analyzing the likelihood dynamics of diffusion SDE, for instance (Huang et al., 2021; Song et al., 2021a; Kingma et al., 2021). These works provide a continuous version of ELBO for diffusion SDEs by different techniques. In this section, we follow

the analysis of (Huang et al., 2021) that adopts a Feymann-Kac representation on $p_T$ then using the Girsanov change of measure theorem to obtain a trainable ELBO.

Here we mainly consider when $g^2(t) \leq 2\beta(t)$, DiffFlow can also be seen as a mixed particle dynamics between SDMs and GANs. Recall in this case, DiffFlow can be rewritten as,

$$dX_t = \underbrace{\left[ \frac{g^2(t)}{2} \nabla \log q(u(t)X_t; \sigma(t)) \right] dt + \sqrt{g^2(t) - \lambda^2(t)} dW_t}_{SDMs} + \underbrace{\left[ \left( \beta(t) - \frac{g^2(t)}{2} \right) \nabla \log \frac{q(u(t)X_t; \sigma(t))}{p_t(X_t)} + f(X_t, t) \right] dt}_{Regularized \ GANs} .$$

Given a time-indexed discriminator $d_{\theta_D^t}(x, t) : \mathbb{R}^d \times [0, T] \to \mathbb{R}^d$ using logistic regression

$$d_{\theta_D^t}(x, t) \approx \log \frac{q(u(t)X_t; \sigma(t))}{p_t(X_t)}$$

and a score network $s_{\theta_c^t}(x, t) : \mathbb{R}^d \times [0, T] \to \mathbb{R}^d$ using score mathcing

$$s_{\theta_c^t}(x, t) \approx \nabla \log q(u(t)X_t; \sigma(t)) .$$

Then the approximated process is given by the following neural SDE:

$$dX_t = \left[ \frac{g^2(t)}{2} s_{\theta_c^t}(X_t, t) \right] dt + \sqrt{g^2(t) - \lambda^2(t)} dW_t + \left[ \left( \beta(t) - \frac{g^2(t)}{2} \right) \nabla d_{\theta_D^t}(X_t, t) + f(X_t, t) \right] dt . \quad (62)$$

We need to answer the question of how to train the score networks $s_{\theta_c^t}(x, t)$ and the time-indexed discriminator $d_{\theta_D^t}(x, t)$ that optimizes the ELBO of the likelihood $\log p_T(x)$.

Following the analysis of (Huang et al., 2021), in order to obtain a general connection between maximal likelihood estimation and neural network training, we need to apply the Girsanov formula to obtain a trainable ELBO for the likelihood of the terminal marginal density. Before we introduce our main theorem, we need the following two well-known results from stochastic calculus. The first one is Feymann-Kac Formula, adapted from Theorem 7.6 in (Karatzas et al., 1991).

**Lemma 3** (Feymann-Kac Formula). *Suppose $u(t, x) : [0, T] \times \mathbb{R}^d \to \mathbb{R}$ is of class $C^{1,2}([0, T] \times \mathbb{R}^d])$ and satisfies the following PDE:*

$$\frac{\partial u(t, x)}{\partial t} + c(x, t)u(t, x) + \frac{\sigma(t)^2}{2} \nabla \cdot \nabla u(t, x) + b(t, x) \cdot \nabla u(t, x) = 0 \quad (63)$$

*with terminal condition $u(T, x) = u_T(x)$. If $u(t, x)$ satisfies the polynomial growth condition*

$$\max_{0 \leq t \leq T} |u(t, x)| \leq M(1 + \|x\|^{2\mu}), \quad x \in \mathbb{R}^d \quad (64)$$

*for some $M > 0$ and $\mu \geq 1$. Then $u(t, x)$ admits the following stochastic representation*

$$u(t, x) = \mathbb{E} \left[ u_T(X_T) \exp \left( \int_t^T c(X_s, s) ds \right) \middle| X_t = x \right] \quad (65)$$

*where $\{X_s\}_{t \leq s \leq T}$ solves the following SDE with initial $X_t = x$,*

$$dX_s = b(t, X_s) dt + \sigma(t) dW_t . \quad (66)$$

Then we need the well-known Girsanov Theorem to measure the deviation of path measures.

**Lemma 4** (Girsanov Formula, Theorem 8.6.3 in (Oksendal, 2013)). *Let $(\Omega, \mathcal{F}, \mathbb{P})$ be the underlying probability space for which $W_s$ is a Brownian motion. Let $\widetilde{W}_s$ be an ito process solving*

$$d\widetilde{W}_s = a(\omega, s) ds + dW_s \quad (67)$$

*for $\omega \in \Omega$, $0 \leq s \leq T$ and $\widetilde{W}_0 = 0$ and $a(\omega, s)$ satisfies the Novikov's condition, i.e.,*

$$\mathbb{E} \left[ \exp \left( \frac{1}{2} \int_0^T a^2(\omega, s) ds \right) \right] < \infty .$$

*Then $\widetilde{W}_s$ is a Brownian motion w.r.t. $\mathbb{Q}$ determined by*

$$\log \frac{d\mathbb{P}}{d\mathbb{Q}}(\omega) = \int_0^T a(\omega, s) \cdot dW_s + \frac{1}{2} \int_0^T \|a(\omega, s)\|^2 ds . \quad (68)$$

With the above two key lemmas, we are able to derive our main theorem.

**Theorem 3** (Continuous ELBO of DiffFlow). *Let $\{\hat{x}(t)\}_{t\in[0,T]}$ be a stochastic processes defined by (62) with initial distribution $\hat{x}(0) \sim q_0(x)$. The marginal distribution of $\hat{x}(t)$ is denoted by $q_t(x)$. Then the log-likelihood of the terminal marginal distribution has the following lower bound,*

$$\log q_T(x) \geq \mathbb{E}_{Y_T}\left[\log q_0(Y_T)\Big|Y_0 = x\right] + \frac{1}{2}\int_0^T \sigma^2(T-s)\mathbb{E}_{Y_s|Y_0=x}\left[\|\nabla \log p(Y_s|Y_0 = x)\|_2^2\right]ds$$

$$-\frac{1}{2}\int_0^T \mathbb{E}_{Y_s|Y_0=x}\left[\left\|\frac{c(Y_s, T-s; \theta_s)}{\sigma(T-s)} - \sigma(T-s)\nabla \log p(Y_s|Y_0 = x)\right\|_2^2\right]ds . \tag{69}$$

*where*

$$dY_s = \sigma(T-s)d\widetilde{W}_s ,$$

*and*

$$c(x, t; \theta_t) = f(x, t) + \frac{g^2(t)}{2}s_{\theta_c^t}(x, t) + \left(\beta(t) - \frac{g^2(t)}{2}\right)\nabla d_{\theta_D^t}(x, t) ,$$

*and*

$$\sigma^2(t) = g^2(t) - \lambda^2(t) .$$

*Proof.* By Fokker-Planck equation, we have

$$\frac{\partial q_t(x)}{\partial t} + \nabla \cdot c(x, t; \theta_t)q_t(x) + c(x, t; \theta_t) \cdot \nabla q_t(x) + \frac{\sigma^2(t)}{2}\nabla \cdot \nabla q_t(x) = 0 \tag{70}$$

where

$$c(x, t; \theta_t) = f(x, t) + \frac{g^2(t)}{2}s_{\theta_c^t}(x, t) + \left(\beta(t) - \frac{g^2(t)}{2}\right)\nabla d_{\theta_D^t}(x, t) ,$$

and

$$\sigma^2(t) = g^2(t) - \lambda^2(t) .$$

Let the time-reversal distribution $v_t(x) = q_{T-t}(x)$ for $0 \leq t \leq T$, then $v_t(x)$ satisfies the following PDE,

$$\frac{\partial v_t(x)}{\partial t} - \nabla \cdot c(x, T-t; \theta_s)v_t(x) - c(x, T-t; \theta_s) \cdot \nabla v_t(x) - \frac{\sigma^2(T-t)}{2}\nabla \cdot \nabla v_t(x) = 0 .$$

By Feymann-Kac formula, we have

$$q_T(x) = v_0(x) = \mathbb{E}\left[q_0(Y_T)\exp\left(-\int_0^T \nabla \cdot c(Y_s, T-s; \theta_s)ds\right)\Big|Y_0 = x\right] \tag{71}$$

where $Y_s$ is a diffusion process solving

$$dY_s = -c(X_s, T-s; \theta_s)ds + \sigma(T-s)dW_s . \tag{72}$$

By Jensen's Inequality,

$$\log q_T(x) = \log \mathbb{E}_{\mathbb{Q}}\left[\frac{d\mathbb{P}}{d\mathbb{Q}}q_0(Y_T)\exp\left(-\int_0^T \nabla \cdot c(Y_s, T-s; \theta_s)ds\right)\Big|Y_0 = x\right]$$

$$\geq \mathbb{E}_{\mathbb{Q}}\left[\log \frac{d\mathbb{P}}{d\mathbb{Q}} + \log q_0(Y_T) - \int_0^T \nabla \cdot c(Y_s, T-s; \theta_s)ds\Big|Y_0 = x\right] . \tag{73}$$

Now, if we choose

$$d\widetilde{W}_s = a(\omega, s)ds + dW_s \tag{74}$$

and $\mathbb{Q}$ as

$$\log \frac{d\mathbb{P}}{d\mathbb{Q}}(\omega)$$

$$= \int_0^T a(\omega, s) \cdot dW_s + \frac{1}{2} \int_0^T \|a(\omega, s)\|^2 ds$$

$$= \int_0^T a(\omega, s) \cdot (d\widetilde{W}_s - a(\omega, s)ds) + \frac{1}{2} \int_0^T \|a(\omega, s)\|^2 ds$$

$$= \int_0^T a(\omega, s) \cdot d\widetilde{W}_s - \frac{1}{2} \int_0^T \|a(\omega, s)\|^2 ds \tag{75}$$

Then $d\widetilde{W}_s$ is Brownian motion under $\mathbb{Q}$ measure and

$$\log q_T(x)$$

$$\geq \mathbb{E}_{\mathbb{Q}} \left[ \int_0^T a(\omega, s) \cdot d\widetilde{W}_s - \frac{1}{2} \int_0^T \|a(\omega, s)\|^2 ds + \log q_0(Y_T) - \int_0^T \nabla \cdot c(Y_s, T - s; \theta_s) ds \middle| Y_0 = x \right]$$

$$= \mathbb{E}_{\mathbb{Q}} \left[ -\frac{1}{2} \int_0^T \|a(\omega, s)\|^2 ds + \log q_0(Y_T) - \int_0^T \nabla \cdot c(Y_s, T - s; \theta_s) ds \middle| Y_0 = x \right]$$

$$= \mathbb{E}_{Y_T} \left[ \log q_0(Y_T) \middle| Y_0 = x \right] - \mathbb{E}_{\mathbb{Q}} \left[ \frac{1}{2} \int_0^T \left[ \|a(\omega, s)\|^2 + \nabla \cdot c(Y_s, T - s; \theta_s) \right] ds \middle| Y_0 = x \right]. \tag{76}$$

Furthermore, we have

$$dY_s = -c(Y_s, T - s; \theta_s)ds + \sigma(T - s)dW_s$$

$$= -(c(Y_s, T - s; \theta_s) + \sigma(T - s)a(\omega, s))ds + \sigma(T - s)d\widetilde{W}_s \tag{77}$$

By choosing appropriate $a(\omega, s)$, we can obtain a trainable ELBO. In particular, we choose

$$a(\omega, s) = -c(Y_s, T - s; \theta_s)/\sigma(T - s). \tag{78}$$

Then we have

$$dY_s = \sigma(T - s)d\widetilde{W}_s. \tag{79}$$

and

$$\log q_T(x)$$

$$\geq \mathbb{E}_{Y_T} \left[ \log q_0(Y_T) \middle| Y_0 = x \right] - \frac{1}{2} \int_0^T \mathbb{E}_{Y_s} \left[ \left( \frac{\|c(Y_s, T - s; \theta_s)\|^2}{\sigma^2(T - s)} + \nabla \cdot c(Y_s, T - s; \theta_s) \right) \middle| Y_0 = x \right] ds$$

$$= \mathbb{E}_{Y_T} \left[ \log q_0(Y_T) \middle| Y_0 = x \right] + \frac{1}{2} \int_0^T \sigma^2(T - s) \mathbb{E}_{Y_s|Y_0=x} \left[ \|\nabla \log p(Y_s|Y_0 = x)\|_2^2 \right] ds$$

$$- \frac{1}{2} \int_0^T \mathbb{E}_{Y_s|Y_0=x} \left[ \left\| \frac{c(Y_s, T - s; \theta_s)}{\sigma(T - s)} - \sigma(T - s)\nabla \log p(Y_s|Y_0 = x) \right\|_2^2 \right] ds. \tag{80}$$

Then we can obtain the objective of the maximum likelihood inference by jointly training a weighted composite network $c(x, t; \theta_t) = f(x, t) + \frac{g^2(t)}{2} s_{\theta_c^t}(x, t) + \left( \beta(t) - \frac{g^2(t)}{2} \right) \nabla d_{\theta_D^t}(x, t)$ to be some weighted version of denoising score matching. $\square$

## J ADDITIONAL DISCUSSIONS

### J.0.1 STOCHASTIC LANGEVIN CHURN DYNAMICS (SLCD)

If $g(\cdot)$ to $g^2(t) > 2\beta(t)$, the GAN component would vanish and we would obtain a Langevin-like algorithm described by the following SDE:

$$dX_t = \underbrace{[f(X_t, t) + \beta(t)\nabla \log q(u(t)X_t; \sigma(t))]}_{Regularized\ Score\ Dynamics} dt + \underbrace{\left( \frac{g^2(t)}{2} - \beta(t) \right) \nabla \log p_t(X_t)}_{Denoising} dt + \underbrace{\sqrt{g^2(t) - \lambda^2(t)}dW_t}_{Diffusion}.$$

We name the above SDE dynamics the Stochastic Langevin Churn Dynamics (SLCD) where we borrow the word "churn" from the section 4 of the EDM paper (Karras et al., 2022), which describes a general Langevin-like process of adding and removing noise according to diffusion and score matching respectively. The SDE dynamics described above is to some sense exactly the Langevin-like churn procedure in the equation (7) of (Karras et al., 2022): the particle is first raised by a gaussian noise of standard deviation $\sqrt{g^2(t) - \lambda^2(t)}$ and then admits a deterministic noise decay of standard deviation of $\sqrt{g^2(t) - 2\beta(t)}$. This is where the name Stochastic Langevin Churn Dynamics comes from.

### J.0.2 ANALYTIC CONTINUATION OF DIFFFLOW

From previous analysis, we know that the scaling function $g(\cdot)$ controls the proportion of GANs component and $\lambda(\cdot)$ controls the stochasticity and aligns the noise level among Langevin algorithms, diffusion SDEs, and diffusion ODEs. We will show later that the change of $g(t)$ would not affect the marginal distributions $p_t(x)$ and only $\lambda(\cdot)$ plays a critical role in controlling the stochasticity of particle evolution. Further more, we could extend to $\lambda^2(t) < 0$ to enable more stochasticity than Langevin algorithms: by letting $\widetilde{\lambda}(t) = \sqrt{-1}\lambda(t)$, we can obtain the following analytic continuation of DiffFlow on $\lambda(t)$:

$$dX_t = \left[ \underbrace{f(X_t, t)}_{Regularization} + \underbrace{\beta(t)\nabla \log \frac{q(u(t)X_t; \sigma(t))}{p_t(X_t)}}_{Discriminator} + \underbrace{\frac{g^2(t)}{2}\nabla \log p_t(X_t)}_{Denoising} \right] dt + \underbrace{\sqrt{g^2(t) + \widetilde{\lambda}^2(t)}dW_t}_{Diffusion} .$$

As shown in Figure 1, the analytic continuation area is marked as grey that enables DiffFlow achieves arbitrarily level of stochasticity by controlling $\widetilde{\lambda}(t)$ wisdomly.

## K POTENTIAL DESIGN SPACE AND RELATIONS TO ZHENG ET AL. (2022)

We show that our framework incorporates some recent proposed hybrid algorithms of GAN and diffusion models, for instance, the TDPM (Zheng et al., 2022) as special case. Notice that when $g^2(t) \leq 2\beta(t)$, DiffFlow can also be written as follows: for $t \in [0, T]$

$$dX_t = \underbrace{\left[\frac{g^2(t)}{2}\nabla \log q(u(t)X_t; \sigma(t))\right] dt + \sqrt{g^2(t) - \lambda^2(t)}dW_t}_{SDMs} +$$
$$\underbrace{\left[\left(\beta(t) - \frac{g^2(t)}{2}\right)\nabla \log \frac{q(u(t)X_t; \sigma(t))}{p_t(X_t)} + f(X_t, t)\right] dt}_{Regularized\ GANs} .$$

Our framework enables both hybrid and multi-stage mixtures of GANs and diffusion models. The GANs, a coarse approximation of diffFlow-ODEs, can achieve fast sampling but with lower sample quality; meanwhile, the diffusion models has higher sample quality but need multistep sampling. Towards this reason, we can design a two stage algorithm of GANs and Diffusions Models to achieve such trade-off: we separate the time interval $[0, T] = [0, \tau] \cup [\tau, T]$ for some $0 < \tau < T$. Then

- for $t \in [0, \tau]$, by setting $\sigma(t) \equiv \sigma_{trunc} > 0$, $u(t) \equiv 1$, $f(x, t) \equiv 0$, $g(t), \lambda(t) \equiv 0$, and $\beta(t) \equiv 1$, we obtain the ODE for GAN training;

- for $t \in [\tau, T]$, by setting $u(t) \equiv \exp\left(\frac{1}{2}\int_0^{T-t}\beta(s)ds\right)$, $f(X_t, t) \equiv \frac{1}{2}\beta(T - t)X_t$, $\beta(t, X_t) \equiv \beta(T - t)$, $\lambda(t) \equiv \sqrt{\beta(T - t)}$, $g(t) \equiv \sqrt{2\beta(T - t)}$ and $\sigma(t) \equiv 1 - \exp\left(-\int_0^{T-t}\beta(s)ds\right)$, we obtain the VP-SDE; if we further descretize the VP-SDE with the schedule in ancestor sampling, we recover the original DDPM on the discretization of truncated time interval $[\tau, T]$.

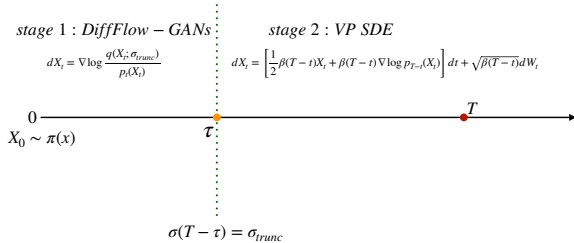

Figure 2: Two Stage Algorithm of DiffFlow.

Notice that the above two-stage algorithm of DiffFlow shares the same spirits with TDPM, and it can exactly match the TDPM algorithm for some particular discretization schedule. Therefore, our framework opens some possibility of design space for generative models to achieve better trade-off between high sample quality and fast sampling speed.

## L  NUMERICAL EXPERIMENTS

### L.1  NUMERICAL IMPLEMENTATION OF DIFFFLOW SDE

We have conducted several numerical examples to validate the feasibility of DiffFlow-SDE algorithm.

We use 3-layer neural network as our function space to approximate the score, where the hidden size is 512 and activation function is Tanh. For DiffFlow SDE, we set $g(t) \equiv 0$, $\beta(t) \equiv 1$, $\sigma(t) \equiv 1 \times 10^{-2}$. The scaling function $f(X_t, t) \equiv c\|X_t\|_2^2$ with $c \equiv 1 \times 10^{-4}$ acts as weight decay. Finally, we consider an additional diffusion term with analytic continuation form of variance $\tilde{\lambda}^2(t) \equiv 1 \times 10^{-4}$. For dataset, we consider (1) Gaussian mixture distribution with 8 centers; (2) two moon dataset, which are multi-modal distributions. It is clear that SFGO algorithm would not suffer from mode collapse and is able to regenerate samples from the approximated true distribution.

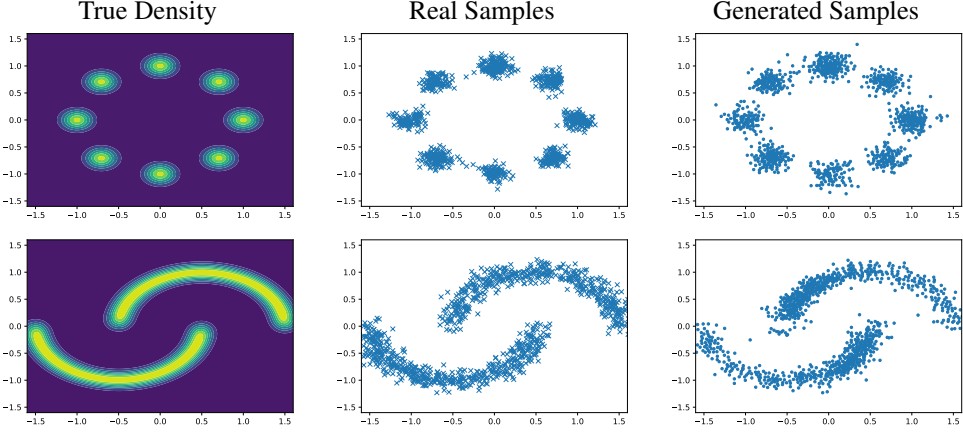

Figure 3: Numerical results for DiffFlow algorithm

### L.2  ON THE EQUIVALENCE BETWEEN DIFFFLOWODE-GAN AND VANILLA DCGAN

By modifying code of DCGAN according to algorithm DiffFlow-GANs in Appendix B, and further train it on MNIST dataset, we conclude that these two algorithms are equivalent up to numerical precision with same random seed.

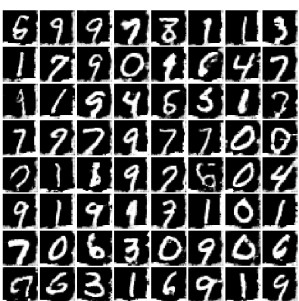

Figure 4: Numerical results for Vanilla DCGAN and DiffFlowODE-GAN on MNIST

## L.3 TRADE-OFF BETWEEN HIGH SAMPLE QUALITY AND FAST SAMPLING SPEED

Since our framework incorporate TDPM as special case, this claim can be justified by reproducing experiments in TDPM (Zheng et al. (2022)).

