# OpenReview forum: "DiffFlow: A Unified SDE for Score-Based Diffusion Models and Generative Adversarial Networks"
_ICLR.cc/2024/Conference — Submitted to ICLR 2024_

### Official Review · Reviewer_WfjL · 2023-10-28

**Soundness:** 4 excellent
**Presentation:** 4 excellent
**Contribution:** 3 good
**Rating:** 6
**Confidence:** 3

**Summary:**

The goal of this paper is to unify the benefits of GANs and Diffusion models into a single model. In particular, GANs are efficient samplers, requiring only one pass of a neural network whereas diffusion models require multiple steps of denoising. Thus, this work derives a stochastic differential equation that encompasses the unification. In particular, there are weighting parameters that yield a smooth transition between GANs and diffusion models. Asymptotic convergence guarantees are also provided along with showing explicit instantiations such as TDPM as special cases.

**Strengths:**

- This paper provides an elegant unification of diffusion models and generative adversarial networks, which are two dominating methods for deep generative modelling, both of which are of great practical significance.
- The unification is more than just a simple weighted combination as exemplified by the marginal preserving property.
- Existing frameworks are demonstrated to be instantiations of this unification such as TDPM, VP SDE, DDPM etc. attesting to the generality of the claims made.
- Convergence guarantees are then given.

**Weaknesses:**

- It seems the instantiation to GANs is given via the coarse approximation which is not as direct and also is based on the density ratio. Some more popular GAN methods do not admit a density ratio perspective such as IPM-based GANs such as Wasserstein GAN, etc.

Minor weaknesses:
- Convergence rates are not given and only an asymptotic guarantee. While this would be difficult to achieve in general, it would be interesting to see how the weighting parameters play a role in the convergence rate.
- There are no experimental studies on more novel / unique choices of weighting parameters. Due to the space requirements however, this is understandable.

**Questions:**

- Is there a way to consider more general GAN frameworks beyond the vanilla GAN? such as Wasserstein GANs or non-saturating GANs.
- There are other methods that use discriminators with diffusion models as a refining process [1]. Can you comment on how such a method relates to the unification provided in this paper?
- Is there any comment you can provide about how different weighting parameters will affect convergence?
- Can you provide more concrete guidance for practitioners willing to use this method and how the weighting parameters inform various choices in practice for different domains?

[1] Kim, Dongjun, et al. "Refining generative process with discriminator guidance in score-based diffusion models." arXiv preprint arXiv:2211.17091 (2022).

---

> ### Author Response · Authors · 2023-11-23
> **Some arguments are defered in appendix.**
>
> Thanks for your positive feedback. It seems that you missed some parts of our arguments such as non-saturating GANs and explicit convergence rate analysis in appendix. The folllowings are point-by-point response to your concerns.
>
> C1:  It seems the instantiation to GANs is given via the coarse approximation which is not as direct and also is based on the density ratio. Some more popular GAN methods do not admit a density ratio perspective such as IPM-based GANs such as Wasserstein GAN, etc.
>
> R1: Yes, you are correct. As pointed out by reviewer Pk86,  the unifying framework only encompasses derivations of the vanilla GAN and non-saturating GANs from Goodfellow et al. (2014), and not all GANs (like Wasserstein GANs, for example). We explicitly state this limitations starting from abstract.
>
> C2: Convergence rates are not given and only an asymptotic guarantee. While this would be difficult to achieve in general, it would be interesting to see how the weighting parameters play a role in the convergence rate.
>
> R2: Due to space limitations, we deferred the proof of explicit convergence rate in Appendix H and discuss how the noise weighting parameters play a role in the convergence rate.
>
> C3: There are no experimental studies on more novel / unique choices of weighting parameters.
>
> R3:  In the revised version, we added some preliminary experiments on the implementation of GANs in the DiffFlow framework. We also have some preliminary implementation on DiffFlow with score-matching by separate networks. We would be grateful if you could further point out additional experiments to strengthen the theory of current paper.
>
> ---------
>
> Q1: Is there a way to consider more general GAN frameworks beyond the vanilla GAN? such as Wasserstein GANs or non-saturating GANs.
>
> A1:  Yes. We have considered both the vanilla GANs and non-saturating GANs in Appendix B. We also discuss how non-saturating GANs avoid gradient vanishing from the perspective of particle gradient optimization.
>
> Q2: There are other methods that use discriminators with diffusion models as a refining process [1]. Can you comment on how such a method relates to the unification provided in this paper?
>
> A2: [1] has fundamental difference from our current framework: given a \textbf{fixed} pretrained score network $\nabla \log p_{model}(x_t, t)$ that may deviates from real data distribution, the authors train an additional time-dependent discriminator $d(x_t, t)$ that discriminates between the real data and generated data at time step $t$ and correct the score network by $$\nabla \log p_{new}(x_t, t) = \nabla \log p_{model}(x_t, t) + \nabla \log \frac{d(x_t, t)}{1-d(x_t, t)}~.$$
>
> In our framework, the score network and the discriminator network should be trained \textbf{jointly}. Furthermore, the noise in the discriminator should align with the diffusion process. Contrastly in [1], there are no noise added on the target distribution $q(x)$ during the training of discriminator. We have added this discussion in Appendix C in the revised submission.
>
>
> Q3: Is there any comment you can provide about how different weighting parameters will affect convergence?
>
> A3: We have discussed this point at the end of Appendix H. In particular, we discuss how regularization and noise weighting on the smoothed target distribution could affect the convergence rate. The general principle is to ensure the isoperimetric property of the target along the optimization path.
>
>
>
> Q4: Can you provide more concrete guidance for practitioners willing to use this method and how the weighting parameters inform various choices in practice for different domains?
>
> A4: One message conveyed by unifying SDE of GANs and diffusion models is to provide a theoretic grounded trade-off between sampling quality and sampling speed, as demonstrated empirically in previous work of TDPM (see table 1,2, and 3 in [2]).
>
> In terms of the choice of weighting parameters, more empirical and theoretic investigations are needed. A preliminary theoretic analysis shows (discussed in appendix H) that the explicit weighted noise smoothing on the target distribution would benefit for the training and sampling process.
>
>
>
> We hope the above response could address your concerns.
>
>
> References:
>
> [1] Kim, Dongjun, et al. "Refining generative process with discriminator guidance in score-based diffusion models." arXiv preprint arXiv:2211.17091 (2022).
>
> [2] Zheng, Huangjie, et al. "Truncated Diffusion Probabilistic Models and Diffusion-based Adversarial Auto-Encoders." The Eleventh International Conference on Learning Representations. 2022.

---

### Official Review · Reviewer_G388 · 2023-10-30

**Soundness:** 4 excellent
**Presentation:** 4 excellent
**Contribution:** 3 good
**Rating:** 5
**Confidence:** 3

**Summary:**

They proposed Discriminator Denoising Diffusion Flow (DiffFlow), which unifiies the explicit generative model and implicit generative model. They theoretically showed that single noise-level Langevin dynamics, score-based diffusion models, and generative adversarial networks can be expressed with the general framework they propose.

**Strengths:**

Originality: They proposed an object function that theoretically integrates GANs and SDMs.

Quality: Written in detail so that even unfamiliar readers can understand.

Clarity: Very well written and easy to read.

Significance: While there have been many studies attempting to integrate GANs and SDMs, theoretical explanations have been lacking, but this paper's theoretical part is very rich.

**Weaknesses:**

It is true that it was expressed very well in theory, but it failed to show the experimental results. If the object of the proposed DiffFlow model was truly an integration of GANs and SDMs, confirmation was needed through experimental results.

**Questions:**

Are there any experimental results that can be shown through this object?

---

> ### Author Response · Authors · 2023-11-23
> **Experiments are added**
>
> Thanks for your constructive comments. We have added experiments about DiffFlow in Appendix L. In particular, the experiments show:
>
> 1. How vanilla GANs can be linked and incorporated into our framework by simply modifying lines of codes from a simple DCGAN.
>
> 2. We also approximate the DiffFlow SDE by neural networks and score matching separately for each discretization step and experiments synthetic datasets to validate the feasibility of DiffFlow algorithm.
>
> Our work is theoretic oriented, and the goal is to give theoretic justifications to existing phenomena and empirical observations, rather than designing models to achieve SOTA performance.

---

### Official Review · Reviewer_WXNS · 2023-10-30

**Soundness:** 3 good
**Presentation:** 2 fair
**Contribution:** 3 good
**Rating:** 6
**Confidence:** 3

**Summary:**

This work provides a unfied formulation of the data generation process of GAN and score-based diffusion model using a SDE, named DiffFlow; variants of score-based diffusion model, langevin algorithm, and GAN simply correspond to specific choice of weighting function presents in DiffFlow. Moreover, a big contribution of this DiffFlow framework is that it naturally leads to several new algorithms that can potentially obtain the advantage of both GAN (fast sampling) and diffusion models (exact likelihood inference and high-quality samples). Finally, several theoretical results are provided to show the convergence of the marginal distributions of the SDE.

**Strengths:**

- The formulation and motivation of the unified SDE is clear.

- The proposed DiffFlow formulation leads to a natural design of DiffFlow-GAN algorithm; based on the discussion provided by the authors in appendix C, this formulation also offers explanations and solutions on the instability of vanilla GAN trainning.

- This work contain an extensive set of theoretical analysis to the SDE.

**Weaknesses:**

- Presentation: Even though the overall formulation of DiffFlow is clear, the reasoning of why such formulation is useful requires is not comprehensive. In particular,  the presentation about the diffusion-GAN algorithm and convergence analysis of DiffFlow is not well structured; many insightful discussions about the trianing of GAN, psudocode of the actual algorithm, further theoretical results, and etc. are deferred to appendix, making readers questioning the real contribution of the proposed algorithm and theory.

Although I understand that the current organization of materials are largely due to the page limiation, it is possible to make the presentation more complete by emphasizing on important contributions.  In my prespective, this work can benefit siginificantly from the following strategies: (1) rearrange the material in Section 3 heavily, e.g, move the majority of 3.2 into the appendix, and dedicate more space for the Diffusion-GAN algorithm; (2) It might be better to just state the informal versions and offer intuitive explanations of major theorems in maintext without getting into details about technical lemmas and assumption, and defer the formal statements and conditions to appendix.


- Lack of experiments: it is generally not fair to ask for experiments on theory-oriented paper. However, a major selling point of such formulation is that it leads to a natural design of the diffusion-GAN method. Pariticularly, given the extensive discussion of the advantage of DiffFlow over vinilla GAN in appendix C, it is important to provide empirical verification on the performance of DiffFlow-GAN.

**Questions:**

None.

---

> ### Author Response · Authors · 2023-11-23
> **Rearrange Section $3$ and add experiments in Appendix L**
>
> Thanks for your detailed comments on the presentation of the paper. We have re-arranged Section $3$, and in particular Section $3.2$ according to your advise. Furthermore, we added several experiments about DiffFlow-GAN to justify its feasibility in appendix L.

---

### Official Review · Reviewer_Pk86 · 2023-10-31

**Soundness:** 1 poor
**Presentation:** 2 fair
**Contribution:** 3 good
**Rating:** 3
**Confidence:** 4

**Summary:**

This paper introduces a unifying framework for score-based diffusion models and GANs. It relies on a stochastic differential equation, called DiffFlow, that incorporates the gradient of the log ratio between a noisy data distribution and the generated distribution. This term being associated to an optimal vanilla GAN discriminator, DiffFlow describes a discriminator-guided dynamics, assimilated to GANs in the paper. DiffFlow also directly encompasses standard score-based models, which can be obtained starting from the GAN dynamics via a smooth interpolation. This unifying framework allows the authors to introduce several new hybrid generative modeling algorithms, and to describe a convergence result for the general dynamics of DiffFlow.

**Strengths:**

The paper tackles a **well motivated and relevant topic**: the links between GANs and score-based diffusion models. Given the usual opposition between both methods in the literature, as highlighted in the paper through the prism of explicit vs implicit models, unifying both of them in a single framework is **interesting** as it opens up potentially fruitful areas of research.

The unifying equation DiffFlow and its dual interpretation between GANs and diffusion models is **novel**, providing a new non-conventional perspective on both models. Some of the introduced algorithms, like Diffusion-GAN, are original as well. To my understanding (up to mathematical details in the appendix that I could not thoroughly check), **the theoretical derivations seem correct**.

Finally, the derivations in the paper are, for the most part, **clearly written** (except Section 3.3, cf. weaknesses). The core ideas and intuitions of the paper are well presented, especially regarding the details of the particular cases of DiffFlow which are well articulated between each other.

**Weaknesses:**

Unfortunately, this paper suffers from important weaknesses that, for some of them, would require significant changes for future acceptance, thus motivating my recommendation of a "reject". I look forward to discussing with the authors and other reviewers on this topic.

### Overclaiming Results

As it is currently written, the paper suffers from **strong overclaiming** of its results. Clarifying the claims and adjusting them so that they are properly supported by theoretical or empirical evidence is a necessity.

Let me recall the stated "research question" of p. 2, to which the authors provide a "positive response".
> Can we develop a unified theoretical framework for GANs and SDMs that allows for a flexible trade-off between high sample quality and fast sampling speed, while enabling exact likelihood inference?

The unifying framework does exist, but there is no element supporting the rest of the claim in the paper. While there are new introduced algorithms, **their properties are not assessed with any experiment**. Moreover, to my understanding, there is no concrete discussion as to why the framework enables exact likelihood inference in its general case (and in particular, for GANs).

Furthermore, the unifying framework only encompasses derivations of the vanilla GAN model from Goodfellow et al. (2014), and not all GANs (like Wasserstein GANs, for example). This should be explicitly stated, starting from the abstract.

### Incomplete and Unclear Link with GANs

An important weakness of the paper and presented framework is the link of DiffFlow with GANs in Section 3.3. While central in the paper's claims and contributions, **the link with GANs is loose and presented in an unclear fashion**.

Firstly, **the development of Section 3.3 is incomplete** to fully understand the presented contribution, as it misses in particular details on how to handle the generator. The development in Appendix Section B should be moved in the main paper for completeness.

Secondly and more importantly, even taking into account Appendix Section B, **the link with GANs remains loose**. The paper does establish that the gradient of the log ratio corresponds to an optimal vanilla GAN discriminator. However, how this can be articulated with GAN generators is left unfinished in the appendix: for example, Equations 27 and 28 only deal with the evolution of generator parameters w.r.t. training time. This does not correspond to Equations 12 or 13 in the main paper, hence to DiffFlow.

This problem is illustrated by Appendix Section K on framing TPDM (Zheng et al., 2022) using the DiffFlow equation. The authors split the generating equation of DiffFlow into two parts: one with discriminator-guided dynamics (assimilated to the first step of TPDM), and one with score-guided dynamics (the second step of TPDM). Yet, in TPDM, the first step is not a discriminator-guided dynamics, but simply a forward pass through a generator, which is not encompassed in the proposed framework.

### Overclaiming Novelty

While the theoretical results seem correct, their presentation is flawed as **it lacks contextualization w.r.t. already existing works**, making them appear more innovative than they actually are. I detail this issue in the following.
- Proposition 1, to my understanding, is similar to results already obtained by Song et al. (2021), who already needed to compute the variance of the marginal distributions.
- The noise-corrupting strategies for Vanilla GANs mentioned in Appendix Sections B and C are strongly linked to preexisting works leveraging noise to regularize the discriminator, e.g. instance noise (Sønderby et al., 2017) and diffusion GANs (Wang et al., 2023).
- Proposition 2 is a direct consequence of the Fokker-Planck equation (see e.g. Jordan et al., 1998). The result may be lesser known in the generative modeling community, but, given the lack of technical novelty, this should be explicitly stated.
- By Proposition 2, I believe that the convergence results (minimization of the KL and Theorem 1) are direct consequences of the fact that the studied dynamics is equivalent to the well known Langevin dynamics; cf. Jordan et al. (1998) and the extensive literature on this topic.

Jordan et al. The variational formulation of the Fokker-Planck equation. SIAM Journal on Mathematical Analysis, 1998.\
Sønderby et al. Amortised MAP inference for image super-resolution. ICLR 2017.\
Wang et al. Diffusion-GAN: Training GANs with Diffusion. ICLR 2023.

### Minor Issues

Less importantly than the above weaknesses, the paper could benefit from further polishing to improve its factualness and readability.
- In the abstract and later in the paper, the authors state that DiffFlow describes "the learning dynamics of both SDMs and GANs". This is incorrect since it does not describe the learning dynamics of SDMs, instead it does describe their generation process.
- It is not clear to me how Remark 1 makes the convergence result of Theorem 1 directly applicable to the general DiffFlow equation.
- The organization of the paper is difficult to follow as some of the main contributions listed in the introduction are relegated to the appendix. All contributions should clearly appear in the main paper. The appendix should also appear in the same file as the main paper.
- The authors state in Section 2.2 that "the training dynamics of GANs are unstable due to the high non-convexity of the generator and discriminator". Explaining instabilities in the training dynamics of GANs remains an open problem, so this statement should be supported with a reference.
- It is not clear how the drift term $f$ can be assimilated to regularization / weight decay.
- It seems that the first two equations of p. 7 are the same.
- Remarks on the form:
  - subsubsection 3.3.1 should not exist as it is the only one of its subsection;
  - all equations should be numbered;
  - the acronym SLCD is not explained in the main paper;
  - the paper Song et al., marked as 2020b in the paper, was actually published in 2021;
  - the name diffusion GANs is already taken by Wang et al. (2023), cf. the "novelty" section of this review.

### Post-Rebuttal

I thank the authors for their response. This response was posted very close to the end of the discussion period, so I will not be able to discuss it with the authors. Nonetheless, I would like to state how and why their response does not affect my recommendation.

#### Overclaiming Results

I acknowledge that the authors clarified their scope on GAN models and removed claim (iii). However, I still contest claim (ii). No part of the introduced framework provides fast sampling, as the generator is left out of the discussion; cf. the next section.


#### Incomplete and Unclear Link with GANs

The authors did not address my concern. The paper assimilates the inference of a GAN (pushforward generator) with its training dynamics (related to the discriminator-guided particle dynamics as already noted by Monoflow, but which cannot be seen as fast sampling) to support their claims. This assimilation is misleading and leads to an incorrect interpretation of TPDM as following a single inference dynamics.

Moreover, the link with GANs remains incomplete both in the main paper and in the appendix. If this link is already supported by Monoflow, I would suggest the authors to precisely describe this link in the main paper instead relying on an incomplete discussion in the appendix. In the current state of the paper, the link remains loose and its articulation with Monoflow is unclear. More generally, the paper relies too much on its appendix; the main paper should be more self-contained.


#### Overclaiming Novelty

The authors acknowledged parts of my concerns regarding overclaiming of novelty. However, they maintained their claim on Langevin convergence, which I have to contest. Langevin dynamics have been extensively studied outside the scope of generative modeling, and, to the best of my knowledge and without further clarification, the presented result is a straightforward application of the literature in Langevin sampling; see for example Proposition 1 by Durmus & Mouline (2018). The smoothing of the data distribution to ensure convergence already existed in the first iterations of score-based diffusion models as well. Furthermore, it is still unclear how this convergence results can be easily extended to more general dynamics; the added comment in the revision should be more explicit.

Overall, the novelty of the theoretical results remains limited. This is not an issue per se but, given that the unification framework lacks substance (cf. section above), this limits the significance of the presented contribution.

Durmus & Mouline. High-dimensional Bayesian inference via the Unadjusted Langevin Algorithm. Bernoulli 25, 2019.

**Questions:**

Cf. the *Weaknesses* part of the review for questions related to paper improvements.

Without consequence on my opinion of the paper, for future versions, I would suggest the authors to discuss the differences with the contemporaneous work of Franceschi et al. (2023) as both this paper and their work have overlapping contributions.

Franceschi et al. Unifying GANs and Score-Based Diffusion as Generative Particle Models. arXiv, 2023.

---

> ### Author Response · Authors · 2023-11-23
> **Thanks for your detailed and constructive comments. (1/3)**
>
> The authors would like to express their sincere gratitude on your detailed and constructive comments. Our paper would improve significantly based on your advice.  Below, we provide a point-by-point response to your concerns and revise/rearrange the paper accordingly.  Kindly note that the revised part is marked as red in the revised submission.
>
> Part 1: Revision on overclaimed results.
>
> C1-2: ...the paper suffers from strong overclaiming of its results...The unifying framework does exist, but there is no element supporting the rest of the claim in the paper. While there are new introduced algorithms, their properties are not assessed with any experiment.
>  ...there is no concrete discussion as to why the framework enables exact likelihood inference in its general case (and in particular, for GANs).
>
>
> R1-2: This work claimed three contributions: i) a unified SDE for GANs and SDMs;  ii) a flexible trade-off between high sample quality and fast sampling speed; iii) exact likelihood inference. While the claims ii) and iii) seem vague in the original version, we argue that these two claims can be supported based on extensive related works:
>
> Claim ii)  a flexible trade-off between high sample quality and fast sampling speed. This idea was initially introduced by TDPM [1] and our work is partially motivated from it. [1] combines the generation process of GANs and diffusion models that enables flexible trade-off between sample quality (FID) and sampling speed (NFE). The authors did extensive experiments on various datasets to validate this idea. While the idea of [1] is promising, the main focus is on empirical validations of compatibility between GANs and diffusion models that enables such a trade-off. Our work give a strict math formulation that justifies this idea theoretically. The unified framework incorporates the TDPM as special case (See R6 for more details), and does have the ability to achieve a flexible trade-off between high sample quality and fast sampling speed.
>
> Claim iii) exact likelihood inference. This part is moved to appendix I.  It has been well investigated in the literature of diffusion models to derive the maximal likelihood training objective for diffusion SDE/ODE that enables likelihood estimation (for example see [2, 3, 4]). The core idea is to derive a trainable continuous-time ELBO for diffusion SDE. For the case of GANs, if we add small gaussian noise in the particle dynamics, a continuous-time ELBO can be derived that enables likelihood estimation. We add more discussion about this point in the appendix I of revised submission.
>
> Since claim iii) is not the main argument in the paper and the techniques adopted are mature in the literature, we only claim (i) and (ii) in the revised version:
>
>
> \emph{Can we obtain a unified theoretic framework for GANs and SDMs that enables a flexible trade-off between high sample quality and fast sampling speed?}
>
>
>
> We hope this adjustment of claims could eliminate the confusion raised by reviewers and readers.
>
>
> C3: ...the unifying framework only encompasses derivations of the vanilla GAN model from Goodfellow et al. (2014), and not all GANs (like Wasserstein GANs, for example). This should be explicitly stated, starting from the abstract.
>
> R3: Thanks for your useful advice. We stress this limitation in the abstract and conclusion with red font.

---

> ### Author Response · Authors · 2023-11-23
> **Thanks for your detailed and constructive comments. (2/3)**
>
> Part 2: Link to GANs with experiments and code in Supplementary material
>
> C4-5: Incomplete and Unclear Link with GANs. ... the link with GANs is loose and presented in an unclear fashion in Section 3.3...
>
> ...taking into account Appendix Section B, the link with GANs remains loose..for example, Equations 27 and 28 only deal with the evolution of generator parameters w.r.t. training time. This does not correspond to Equations 12 or 13 in the main paper, hence to DiffFlow.
>
> R4-5: This is a misconception. It is well established in the existing literature [7, 8] that the vanilla GAN or non-saturating GAN dynamics (eqn 27 and 29) can be *exactly* recovered by *one-step distillation* of the discriminator-guided particle dynamics (i.e., equation 12 or 13). The equivalence between the vanilla GAN and discriminator-guided dynamics GANs can be obtained by *rescaling the gradient* of the least square distillation objective in algorithm $1$. In the work of MonoFlow, the authors also discussed the problem of gradient vanishing in vanilla GANs arises from too small rescaled vector fields of discriminator-guided particle dynamics and it fixes the gradient vanishing problem by simply adding a constant in the generator output combined with a monotonic function (section 5.2 of [7]).
>
> In the revision, we add additional discussions about this point in the main paper and the end of appendix C. Due to space limitations and since the connection of DiffODE and vanilla GANs are mature in the literature, we will not move too much content in the main paper.
>
> C6: This problem is illustrated by Appendix Section K on framing TPDM (Zheng et al., 2022) using the DiffFlow equation.  in TPDM, the first step is not a discriminator-guided dynamics, but simply a forward pass through a generator, which is not encompassed in the proposed framework....
>
> R6: As we discussed before, a forward pass through a generator can be trained by both vanilla GANs or *one-step distillation* of the discriminator-guided particle dynamics with rescaled gradients.  These two methods of training the generator are equivalent up to numerical precision.
>
> We implement a simple code in Jupyter notebook to show the equivalence between two methods and further confirm the equivalence between DiffFlow-GANs and vanilla GANs.
>
> -----
>
> Part 3: Revision on claiming of novelty
>
> C7: Overclaiming Novelty....Proposition 1 ... is similar to results already obtained by Song et al. (2021), who already needed to compute the variance of the marginal distributions.
>
> R7: This is a misunderstanding.  Song et al. (2021) compute the variance of conditional marginal distributions (transition kernels) $p_{0t}(x_t|x_0)$; in Proposition $1$, we instead compute the exact density formulation of unconditional marginal distribution $p_t(x)$. While it seems viable to achieve the Proposition $1$ based on the relationship between initial distribution and transition kernels, it needs somewhat tricky math manipulations.
>
> The derivation of Proposition $1$ is straightforward by directly applying exponential transformations and Ito formula on diffusion SDE, which is a basic technique in stochastic calculus. Hence, we did not quote Song et al. (2021) near Proposition 1 in the original submission. In the revised submission, we cite Song et al. (2021) before Proposition 1 and move it to appendix A.
>
>
> C8: The noise-corrupting strategies for Vanilla GANs mentioned in Appendix Sections B and C are strongly linked to preexisting works leveraging noise to regularize the discriminator, e.g. instance noise (Sønderby et al., 2017) and diffusion GANs (Wang et al., 2023).
>
> R8: Thanks for mentioning these related works. We have added discussions and comparisons of instance noise (noise-smoothed KL) and diffusion GANs (Wang et al., 2023) in the revision. Kindly note that the diffusion GAN algorithm from (Wang et al., 2023) is quite different from the diffusion-GAN in our framework: in each discriminator-generator optimization cycle, (Wang et al., 2023) propose to train and combine a series of discriminators that are trained by corrupting instance with different levels of noise. Since the noise schedule is aligned with diffusion models, the author name it diffusion GAN. The diffusion GAN in (Wang et al., 2023) is essentially a GAN-like algorithm.  While in our framework of diffusion GAN, we should jointly train a score network and GAN-flavoured generator and the generation process is guided by both networks. Hence, it is a real combination of GANs and diffusion models.
>
>
> C9: Proposition 2 is a direct consequence of the Fokker-Planck equation (see e.g. Jordan et al., 1998)... this should be explicitly stated.
>
> R9: Thanks for your advice. We explicitly state that Proposition 2 is a direct consequence of the Fokker-Planck equation (or Kolmogorov Forward Equation) in the revision.

---

> ### Author Response · Authors · 2023-11-23
> **Thanks for your detailed and constructive comments. (3/3)**
>
> C10: By Proposition 2, I believe that the convergence results (minimization of the KL and Theorem 1) are direct consequences of the fact that the studied dynamics is equivalent to the well known Langevin dynamics; cf. Jordan et al. (1998) and the extensive literature on this topic.
>
> R10: This is a misunderstanding. While the minimization of KL (decreasing lemma) can be easily obtained by the well-established theorems from Wasserstein gradient flows, the convergence is hard to obtain for a general target $q(x)$. In the standard literature of sampling with Langevin dynamics, it is assumed that $q(x)$ satisfies some property of isoperimetry inequality, such as poincare or logarithmic sobolev inequality. Then, the convergence rate can be explicitly obtained based on the estimation of constants of isoperimetry  inequalities (see a seminar work of [5]).
>
> While in Theorem 1, we did not assume any isoperimetry conditions on the target $q(x)$. We instead show that by explicitly smoothing the the functional objective,  we can obtain an asymptotic convergence of the dynamics of the diffusion GAN.
>
> From a technical perspective, the asymptotic convergence is obtained by constructing a Gaussian-Poincare inequality based on the estimation of the smoothed log density ratio (see Lemma $1$ for details). To the best of our knowledge, this series of techniques to obtain Theorem 1 (eqn37--eqn52 in the original paper) is nontrivial and there is no similar technique adopted in the previous literature.
>
> In appendix H, we discuss in Theorem 2 on how to use standard techniques in the well-established Langevin dynamics literature to obtain a linear convergence analysis of diffusion GAN dynamics under stronger assumptions of the target distribution $q(x)$. We defer Theorem 2 to appendix since it can be easily obtained based on the fact that the studied dynamics is equivalent to the well known Langevin dynamics.
>
> ---
>
> Part 4: Discussing contemporaneous work and polishing the paper.
>
> C11: Less importantly than the above weaknesses, the paper could benefit from further polishing to improve its factualness and readability. ....
>
> R11: The authors would thank again for your patience and time on giving useful comments about the current submission. We have reviewed these comments in detail and polish the revision accordingly.
>
>
> C12: Discuss the contemporaneous work of Franceschi et al. (2023).
>
> R12: The contemporaneous work of Franceschi et al.  proposed the Generative Particle Model (GPM) [6] that unifies GANs and diffusion models. Our work shares the same motivation and idea with Franceschi et al. in the sense that both GANs and diffusion models can be viewed as a particle optimization algorithm.  Our work differs from Franceschi et al. (2023) mainly in two aspects:
>
> i). we propose an explicit unified SDE for both GANs and diffusion models while there is no explicit unified SDE formulation for GPM. Under the GPM framework, GANs and diffusion models has different specific formulations of SDE or ODE.
>
> ii).  the GPM paper provides much more empirical analysis and focus less on theoretic analysis. Our work can be seen as a good complementary to the contemporaneous GPM paper.
>
> ---
>
>
> References:
>
> [1]. Zheng, Huangjie, et al. "Truncated Diffusion Probabilistic Models and Diffusion-based Adversarial Auto-Encoders." The Eleventh International Conference on Learning Representations. 2022.
>
> [2]. Huang, Chin-Wei, Jae Hyun Lim, and Aaron C. Courville. "A variational perspective on diffusion-based generative models and score matching." Advances in Neural Information Processing Systems 34 (2021): 22863-22876.
>
> [3]. Song, Yang, et al. "Maximum likelihood training of score-based diffusion models." Advances in Neural Information Processing Systems 34 (2021): 1415-1428.
>
> [4]. Kingma, Diederik, et al. "Variational diffusion models." Advances in neural information processing systems 34 (2021): 21696-21707.
>
> [5]. Ma, Yi-An, et al. "Sampling can be faster than optimization." Proceedings of the National Academy of Sciences 116.42 (2019): 20881-20885.
>
> [6]. Franceschi, Jean-Yves, et al. "Unifying GANs and Score-Based Diffusion as Generative Particle Models." Advances in Neural Information Processing Systems (23')
>
> [7]. Mingxuan Yi, Zhanxing Zhu, and Song Liu. 2023. MonoFlow: rethinking divergence GANs via the perspective of Wasserstein gradient flows. In Proceedings of the 40th International Conference on Machine Learning (ICML'23).
>
> [8]. Gao, Yuan, et al. "Deep generative learning via variational gradient flow." International Conference on Machine Learning. PMLR, 2019.

---

### Meta-Review · Area_Chair_wAUX · 2023-12-09

**Metareview:**

The authors propose a novel unification of what they call implicit and explicit likelihood generative models (e.g GANs vs e.g. slows) that they call Discriminator Denoising Diffusion Flow (DiffFlow), where the generative model is learned as a weighted combination of scores on real and generated data. The framework encompasses recently proposed hybrid adversarial and diffusion based approaches.

The formulation in the paper starts with the SDE formulation of Song et al 2021 and extends the formulation with a discriminator term (see e.g. 12). The contribution is for the time being purely methodological and theoretical. As also stated in the reviews, this is fine, but both reviewers and this AC agree that the framework comes across as not quite completed. Therefore, it is better to reject the paper in its current shape to allow the authors to finish the framework and submit it to the next conference.

**Justification For Why Not Higher Score:**

It is a very interesting proposal but not quite there for publication yet.

**Justification For Why Not Lower Score:**

None

---

### Decision · Program_Chairs · 2024-01-16

Reject